

# Tropical tropospheric aerosol sources and chemical composition observed at high-altitude in the Bolivian Andes

C. Isabel Moreno[1], Radovan Krejci[2], Jean-Luc Jaffrezo[3], Gaëlle Uzu[3,4], Andrés Alastuey[5], Marcos Andrade[1,6], Valeria Mardóñez[3], Alkuin Maximilian Koenig[3], Diego Aliaga[7], Claudia Mohr[2], Laura Ticona[1], Fernando Velarde[1], Luis Blacutt[1], Ricardo Forno[1], David N. Whiteman[8], Alfred Wiedensohler[9], Patrick Ginot[3,4], Paolo Laj[3,7]

[1] Laboratorio de Física de la Atmósfera, Instituto de Investigaciones Físicas, Universidad Mayor de San Andrés, La Paz, Bolivia

[2] Department of Environmental Science & Bolin Centre of Climate Research, Stockholm University, Stockholm 10691, Sweden

[3] Université Grenoble Alpes, CNRS, IRD, Grenoble INP, Institut des Géosciences de l'Environnement, Grenoble, 38400, France

[4] Institut de Recherche pour le Développement, France

[5] Institute of Environmental Assessment and Water Research (IDAEA) - Consejo Superior de Investigaciones Científicas (CSIC), 08034 Barcelona, Spain

[6] Department of Atmospheric and Oceanic Sciences, University of Maryland, College Park, MD 20742, USA

[7] Institute for Atmospheric and Earth System Research/Physics, Faculty of Science, University of Helsinki, Helsinki, 00014, Finland

[8] Howard University, Washington, DC, USA

[9] Leibniz Institute for Tropospheric Research, Permoserstr. 15, Leipzig, 04318, Germany

*Correspondence to*: Carina Isabel Moreno Rivadeneira (caisa.moreno@chacaltaya.edu.bo)



**Abstract.** The chemical composition of $PM_{10}$ and $PM_{2.5}$ was studied at the summit of Mt. Chacaltaya (5380 masl, lat.-16.346950º, lon. -68.128250º) providing a unique long-term record spanning from December 2011 to March 2020. The chemical composition of aerosol at the Chacaltaya GAW site is representative of the regional background, seasonally affected by biomass burning practices and by nearby anthropogenic emissions from the metropolitan area of La Paz – El Alto. Concentration levels are clearly influenced by seasons with minimum occurring during the wet season (December to March) and maxima occurring during the dry and transition seasons (April to November). Ions, total carbon (EC+OC) and saccharide concentrations range between 558-1785, 384-1120 and 4.3-25.5 ng m$^{-3}$ for bulk $PM_{10}$ and 917-2308, 519-1175 and 3.9-24.1 ng m$^{-3}$ for $PM_{2.5}$, respectively. Such concentrations are overall lower compared to other high-altitude stations around the globe, but higher than Amazonian remote sites (except for OC). For $PM_{10}$, there is dominance of insoluble mineral matter (33-56% of the mass), organic matter (7-34%) and secondary inorganic aerosol (15-26%). Chemical composition profiles were identified for different origins: EC, $NO_3^-$, $NH_4^+$, glucose, $C_2O_4^{-2}$ for the nearby urban and rural areas; OC, EC, $NO_3^-$, $K^+$, acetate, formiate, levoglucosan, some $F^-$ and $Br^-$ for biomass burning; $MeSO_3^-$, $Na^+$, $Mg^{2+}$, $Br^-$ for aged marine emissions from the Pacific Ocean; arabitol, mannitol, $K^+$ for biogenic emissions; $Na^+$, $Ca^{2+}$, $Mg^{2+}$ for soil dust, and $SO_4^{2-}$, $F^-$, and some $Cl^-$ for volcanism. Regional biomass-burning practices influence the soluble fraction of the aerosol particularly between July and September. The organic fraction is present all year round and has both anthropogenic (biomass burning and other combustion sources) and natural (primary and secondary biogenic emissions) origins, with the OC/EC mass ratio being practically constant all year round (10.5±38.9). Peruvian volcanism dominates the $SO_4^{2-}$ concentration since 2014, though it presents a strong temporal variability due to the intermittence of the sources and seasonal changes on the transport patterns. These measurements represent some of the first long-term observations of aerosol chemical composition at a continental high-altitude site in the tropical Southern hemisphere.

## 1 Introduction

Aerosol particles are important climate-forcers. Recent estimates suggest that the impact of these aerosols on global climate falls within the range of -2.0 to –0.6 W m$^{-2}$ (Forster et al., 2021). This impact, known as aerosol effective radiative forcing, is characterized by a high level of spatial and temporal heterogeneity (Szopa et al., 2021). Aerosol chemical composition, being key to determine climate relevant properties of aerosol particles such as hygroscopicity and refractive index, has been recently listed as an aerosol Essential Climate Variable, as defined by the Global Climate Observing System (GCOS) of the Atmosphere Ocean Panel on Climate (WMO, 2022). While aerosol chemical composition can be measured in near real time via mass spectrometry techniques or approximated through a combination of indirect observations, actual determination by chemical analysis of aerosol filters remains the simplest solution for providing a precise and quasi-exhaustive knowledge of aerosol constituents (EMEP Manual for Sampling and Analysis, 2020).

In recent decades, great progress has been made in long-term monitoring of aerosol properties, but many temporal and spatial observational gaps remain. South America is one of the regions with under-sampled Essential Climate Variables. In this region,





aerosol chemical composition is available for urban and suburban locations (e.g. Jorquera and Barraza, 2013; Barbosa, 2014; Jorquera, 2008; Olson et al., 2021; Mardoñez et al., 2022; Custodio et al., 2019) and for intensive measurement campaigns in Brazilian Amazonia (Artaxo et al., 2009; Martin et al., 2010, 2017 et references therein). For urban areas, the traffic-associated

markers are unsurprisingly dominant and each location presents its specificities. However, it is noteworthy that emissions associated with agricultural practices exert a significant influence at a continental scale, influencing the air quality in urban areas located far from the sources (e.g. Longo et al., 2009; Custodio et al., 2019; Martin et al., 2010; Giglio et al., 2013; Mardoñez et al., 2022; Estellano et al., 2008). The studies in Amazonia permitted characterization of biomass combustion sources (e.g. Schkolnik et al., 2005; Fuzzi et al., 2007; Kundu et al., 2010b, a) and natural emissions of the Amazonian

environment (e.g. Graham, 2002; Elbert et al., 2007; Claeys et al., 2004).

For the Andean region, measurements of background aerosol chemical composition are in general lacking. However, a small number of studies of certain sources of atmospheric aerosols at high-altitude are available. Some of them are based on the analysis of ice cores (Correia et al., 2003; de Angelis et al., 2003; Brugger et al., 2019; Hong et al., 2004; Magalhães et al., 2019), but these results have not been translated to an equivalent atmospheric chemical composition. Other studies focus on

the characterization of atmospheric aerosols (e.g. Adams et al., 1977; Bianchi et al., 2021; Scholz et al., 2023; Chauvigné et al., 2019; Van Espen and Adams, 1983) during limited time periods.

The objective of this work is to study aerosol chemical composition based on multi-year observations at the Chacaltaya Global Atmosphere Watch (GAW) station. The aerosol at Chacaltaya exhibits influences from different provenances, including planetary boundary layer air masses influenced by the metropolitan area of La Paz and El Alto, as well as long-range

transported air masses (Aliaga et al., 2021, Wiedensohler et al., 2018, Chauvigné et al., 2019). It is interesting to note that more than four decades ago it was considered a background site with very little anthropogenic influence (Adams et al., 1977; Van Espen and Adams, 1983; Cautreels et al., 1977; Adams et al., 1983). Our study presents the chemical composition of $PM_{10}$ and $PM_{2.5}$ for major soluble ions, saccharides and EC-OC obtained at this site between December 2011 and March 2020. This nine-year record is analyzed in the light of possible source regions and transport characteristics, providing unique information

on the main drivers influencing the regional aerosol composition of the Altiplano and adjacent Andes.

With this analysis, we will complement some knowledge gaps for the Andean region, particularly regarding the seasonality of the aerosol observed at high-altitude. We will also situate the Chacaltaya observations in a global and regional context. Additionally, two specific cases are included. The first, focuses on the influence of volcanism on the site. The second, emphasizes the differences between minimum and maximum planetary boundary layer influences on aerosol chemical

composition at this site.





## 2 Methodology

### 2.1 Location and climatology

The Chacaltaya GAW station (lat. -16.350500º, lon. -68.131389º, 5240 m a.s.l.) is currently the highest site in the GAW network. It is located at Mount Chacaltaya in the Bolivian Andes (Figures 1 and S1), where it shares facilities with the Cosmic
Ray Observatory of the Instituto de Investigaciones Físicas (IIF) of the Physics Department of the Universidad Mayor de San Andrés (UMSA). Chacaltaya station is located some 1400 m above the Altiplano high-plateau (mean elevation of about 3750 m a.s.l.), which is bordered to the west by the Cordillera Occidental, which includes a volcanic arc, and to the east by the Cordillera Oriental. Mt. Chacaltaya lies within the northern segment of the Cordillera Oriental, called the Cordillera Real, which trends SE - NW. The semi-arid Altiplano (grasslands, Puna) incorporates Lake Titicaca. Mt. Chacaltaya is located about
40 km from the nearest part of the lake, and about 15 km north from the metropolitan area of La Paz-El Alto. El Alto (La Paz) is a fast-growing city, with its population having increased from approximately 60,000 (595,000) inhabitants in 1976 to 921,000 (782,000) inhabitants in 2012 (Censos de población y vivienda, 2021). In the Peruvian branch of the Cordillera Occidental, about 350 km from the station, there are two active volcanoes: Sabancaya (5976 m asl) and Ubinas (5672 m asl). North and east of Mt. Chacaltaya, beyond the high Cordillera Real, there is a lower and parallel mountain belt known as the
Subandes (paramo and montane forest locally referred to as the Yungas), and beyond the Subandes lie the Amazonian lowlands. The valleys of the Cordillera Real and the Subandes are often steep and forested, and sparsely populated.

General atmospheric circulation shows a pronounced annual cycle of changing dry and wet seasons, typical of the tropical zone. Regional climatological information is well described in other works (Espinoza et al., 2020; Arias et al., 2021). At Chacaltaya station, the mean atmospheric pressure is 534 hPa about half of the value at sea level, mean temperature around
0ºC and annual precipitation (mostly solid) 865 mm (Perry et al., 2017). Four seasons are defined for this region based on the circulation and precipitation changes: 1. Wet season (monsoon-like season) from December to March, bringing 65-70% of the annual precipitation to Chacaltaya, 2. Wet-to-dry transition season in April, 3. Dry season from May to August, with long periods without precipitation that can span more than 40 consecutive days (Andrade et al., 2017) and finally 4. Wet-to-transition season from September to November, that used to mark the onset of the rainy season that has become quite dry over
the last decades (Espinoza et al., 2019). During the wet season, the station is influenced by air-masses from the SE, E and NE (Aliaga et al., 2021), often channeled through the valleys north of the station (Chauvigné et al., 2019; Perry et al., 2017; Vimeux et al., 2005). In the dry and dry-to-wet seasons, winds predominantly blow over the Altiplano and bring mineral dust to Chacaltaya station along with biomass burning tracers (Chauvigné et al., 2019). Biomass burning emissions are transported to the Altiplano through the Subandes valleys east of the Cordillera Real (Bourgeois et al., 2015; Magalhães et al., 2019). On
an annual basis, less than 10% of air masses arrive in the region directly from the Pacific Ocean (Chauvigné et al., 2019; Perry et al., 2017).

Chacaltaya station is frequently influenced by the atmospheric boundary layer (Collaud Coen et al., 2018). During daytime, the Altiplano is subjected to solar radiation heating (Zaratti and Forno, 2003) resulting in a highly convective planetary



boundary layer (PBL). The diurnal development of the PBL routinely brings air pollutants from the nearby metropolitan area

of La Paz - El Alto to Chacaltaya (Wiedensohler et al., 2018; Chauvigné et al., 2019) all year round and without a clear

seasonality (around 24% of the air masses have a PBL influence independently of the long-range origin, Aliaga et al. 2021).

After sunset, the station is frequently situated within a residual layer (RL) (Figure S10), which acts as a barrier, trapping

pollutants and keeping them confined until late in the evening. Purely free tropospheric (FT) intrusions have been observed

for some very short-time periods during the night (Scholz et al., 2023; Zha et al., 2023), and most of the time, the station is

influenced by a mixture of air-masses from below and above the station (Aliaga et al., 2021).

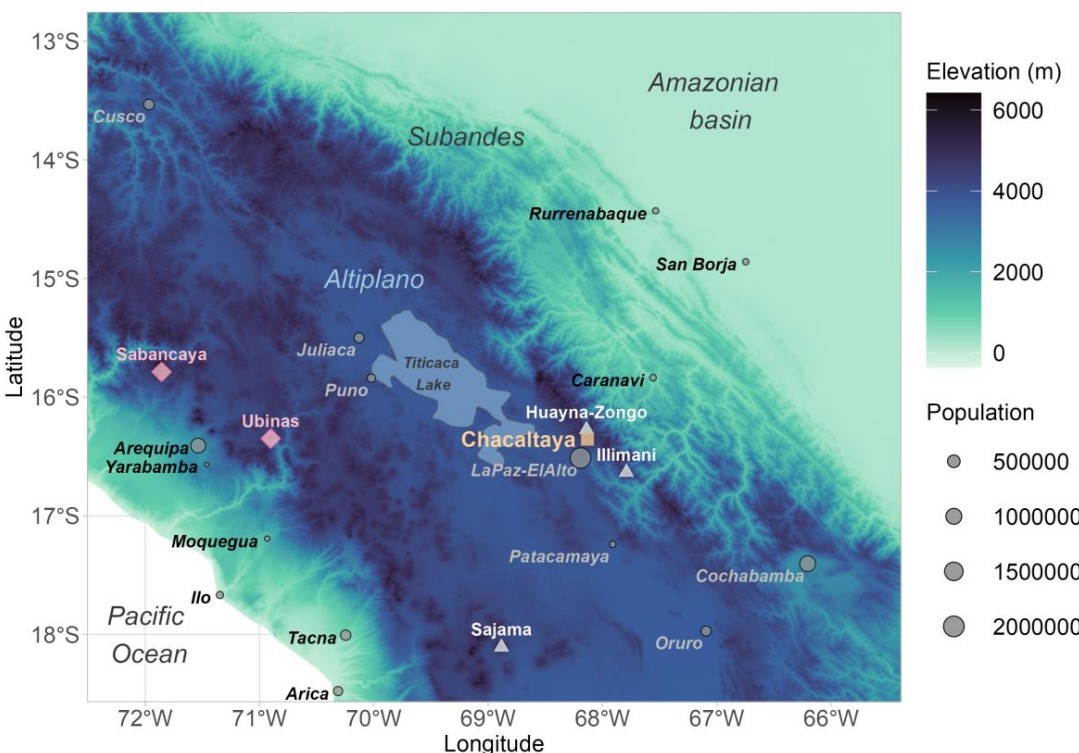

**Figure 1. Location map. Square is the Chacaltaya GAW station, rhombuses are currently active volcanoes, triangles are other mountain sites where ice-core studies were performed, circles are urban sites with population higher than 40000 inhabitants (estimates for 2020 from World Population Review, 2022). Elevation from NOAA National Centers for Environmental Information**

**(2022), Titicaca Lake shape from Natural Earth (2022).**

**2.2 Aerosol sampling**

Aerosol samples were collected on Pallflex pre-baked (at 500-550ºC) quartz filters (Ø=150 mm) using a High Volume Sampler

(HiVol) that was installed in a small concrete hut at the northwestern ridge of Mt Chacaltaya some 120 meters above the main

station building (5380 m a.s.l., lat.-16.346950º, lon. -68.128250º, Figure S1), with few exceptions (very few samples were

taken at the main observatory). The site is accessible only by foot from the main observatory. This HiVol consisted of a

DIGITEL HVS DPM10/30/00 inlet with interchangeable impactor plates for 10 and 2.5 µm aerodynamical sizes coupled to an



Elmo-Rietschle G-BH7 vacuum pump, regulated to work at a nominal flow rate of $30\pm6$ m$^3$.h$^{-1}$ at ambient conditions. The volume of sampled air was obtained from readings of a Honeywell Elster Quantometer QA at the start and the end of the sampling period. Note that given the low atmospheric pressure at Chacaltaya, the sampled volume almost doubles (1.89) the equivalent sampled volume at sea level. Each filter collected was folded, then wrapped in aluminum foil, sealed in a polyethylene plastic bag (sandwich Ziploc bag) and kept at -10ºC prior to shipping to the Institute des Géosciences de l'Environnement (IGE, Grenoble) where the water-soluble fractions (ions and sugars) and EC-OC were analyzed. Samples were sent once a year to the IGE and analyzed up to three years after the sample was taken. Except during transportation, filters were stored at -18ºC.

The vacuum pump was controlled by a timer to take either diurnal, nocturnal or 24-hour samples, and ran for several consecutive days in order to accumulate enough mass for the analysis. Indeed, sampling periods shorter than 24 hours (in total) caused several species to fall below the quantification limit, especially during the core of the wet season. Therefore, later in the experiment, sampling periods of at least seven days were preferred, although some samples extended up to 10 days (intermittently) when it was impossible to access the hut due to bad weather. The night and daytime sampling was tested for several split times in the hope of capturing purely lower free troposphere air which finally proved unsuccessful. Among all the split times tested, the samples taken between 02:00 and 09:00 BOT (UTC-4) were the more likely to capture more LFT episodes, and their complementary samples from 09:00 to 02:00 of the next day would be capturing more periods with BL and RL influence (see section 3.4). Nevertheless, once the diurnal city influence was clearly identified at Chacaltaya (Wiedensohler et al., 2018), nighttime sampling (23:00 to 08:00) was preferred in order to better characterize regional aerosol characteristics, minimizing the city influence. Details on the sampling split times are presented in Table 1.

Until July 1$^{st}$ 2016, the inlet lacked an inner water collection system (Figure S1). As a result, during the wet season, several filters presented marks of water runoff or were drenched. These samples were not analyzed and this led to under-sampling of the rainy season, particularly January. No vaseline was used on the impactor plate, and therefore the samples may have had an influence of particle bouncing off, especially prior to the installation of the inner water collection system which also acts as a barrier for it.

The HiVol sampler worked from December 2011 until March 2020, with PM$_{10}$ samples not overlapping with PM$_{2.5}$. Overall, sampling logistics are challenging and there are five gaps in the data set that can be gathered in three groups: 1. Gaps due to wet samples, difficult conditions to access the sampling hut due to snowfalls, and/or lightning producing power outages at the sampling hut: December 2013 to April 2014, December 2014 to January 2015, and January 2016 to April 2016, 2. Gap due to a broken down flowmeter: August 2017 to December 2018, 3. Gap due to electric line maintenance followed by a period of social unrest that made it impossible to access the station: September 2019 to January 2020. The data series ended in March 2020 due to the Covid-19 lockdown and the theft of the pump while the station remained closed.

In total, 67 field blanks and 242 samples were collected (aside from those lost due to drenching), from which six samples were excluded from the analysis because the flow was too high ($>36$ m$^3$.h$^{-1}$), and 13 because of specific events (San Juan bonfire festivity) or if the sampled volume was small ($<300$ m$^3$). Some of the latter were helpful as study cases for particular conditions



or events but are not presented here. The remaining 223 valid samples are detailed in Table 1. The number of field blanks is very large, which is essential in order to achieve a good contamination control on the data.

**Table 1. Sampling characteristics for High-Volume samples at Chacaltaya station and number of samples per season. PM$_{10}$ was divided into three batches (A,B,C) following important gap periods. N stands for the number of consecutive days for sampling (intermittently for day and night samples). Sampling time (start, end, and duration) is indicated in hours and minutes. Local time (BOT) corresponds to UTC-4.**

| Sample group | Sampling period | N (days) | Type | Sampling hours, local time (start-end) | Wet season (DJFM) | Wet-to-dry transition (A) | Dry season (MJJA) | Dry-to-wet transition (SON) | Total |
|---|---|---|---|---|---|---|---|---|---|
| PM$_{10}$-A | December 2011 to July 2013 | 3 to 7 | Day | 07:00-19:00 | 8 | 1 | 7 | 7 | 59 |
| | | | Night | 19:00-07:00 | 7 | 4 | 12 | 6 | |
| | | | 24H | around 12:00 | 2 | - | 4 | 1 | |
| PM$_{2.5}$ | August 2013 to December 2015 | 5 to 10 | Day | 07:00-19:00\|09:00-02:00\|09:00-23:00 | 4 | 4 | 14 | 10 | 80 |
| | | | Night | 19:00-07:00\|02:00-09:00\|23:00-08:00 | 5 | 2 | 20 | 18 | |
| | | | 24H | around 12:00 | 2 | - | 1 | - | |
| PM$_{10}$-B | April 2016 to August 2017 | 7 to 10 | Day | 09:00-23:00 | - | - | 2 | - | 52 |
| | | | Night | 23:00-08:00 | 11 | 5 | 18 | 9 | |
| | | | 24H | around 12:00 | 3 | - | 3 | 1 | |
| PM$_{10}$- C | February 2019 to March 2020 | 7 | Day | 09:00-23:00 | 1 | 2 | 7 | 1 | 32 |
| | | | Night | 23:00-08:00 | 4 | 2 | 7 | 1 | |
| | | | 24H | around 12:00 | 6 | 0 | 1 | 0 | |

## 2.3 Mass and elemental composition

For nine PM$_{10}$ samples (taken every two months from April 2016 to August 2017), additional analyses were performed at IDAEA (Spain). Gravimetric concentrations of particulate matter were obtained by weighing the filters, prior and post-sampling, after 48h stabilization at 20°C and 50% relative humidity. After gravimetric analysis the concentrations of major and trace elements in particulate matter were determined by following the methodology described elsewhere (Querol et al., 2001; Rodríguez et al., 2011; Ripoll et al., 2015). In short, a fraction from each filter was fully acid digested using a

HNO$_3$:HF:HClO$_4$ mixture. The acidic solution obtained was analyzed by Inductively Coupled Plasma Atomic Emission Spectrometry (ICP-AES) and ICP - Mass Spectrometry (ICP-MS) for the determination of major (Al, Ca, Cu, Fe, K, Mg, Mn, Na, P, and S) and trace elements (Li, Be, Sc, Ti, V, Cr, Mn, Co, Ni, Cu, Zn, Ga, Ge, As, Se, Rb, Sr, Y, Zr, Nb, Mo, Cd, Sn, Sb, Cs, Ba, REE, W, Ti, Pb, Bi, Th, and U), respectively. Two samples were discarded due to instrumental problems (power outage, filter badly set).

The elemental composition was used to calculate the mineral matter (MM) contribution to the samples using Eq. 1:

$$MM = CO_3^{-2} + SiO_2 + Al_2O_3 + P_2O_5 + Fe + Ca + K + Na + Mg + Ti + Mn \ , \tag{1}$$

where SiO2 was estimated from Al and CO32- from Ca (Alastuey et al., 2016).



**2.4 Ions and carbonaceous matter**

Samples were analyzed for their content of soluble ions, elemental and organic carbon and sugar derivatives using sub-sampled fractions of the collection filters. Carbonaceous matter was analyzed using a thermo-optical method on a Sunset Lab analyzer (Birch and Cary, 1996) as described by Aymoz et al. (2007). Carbonaceous matter is subdivided in organic carbon (OC), which is the volatile and non-light-absorbing fraction, and elementary carbon (EC), which is optically absorptive, using the EUSAAR2 protocol (Cavalli et al., 2010). For the analyses of ions and sugars, filter punches (typically of about 10 cm²) were

first extracted into ultrapure water for 20 min in a vortex shaker and then filtered using a 0.22 µm Acrodisc filter. Soluble anions ($F^-$, $Cl^-$, $Br^-$, $NO_3^-$, $SO_4^{2-}$) and cations ($Na^+$, $Li^+$, $NH_4^+$, $K^+$, $Mg^{2+}$, $Ca^{2+}$) were analyzed by ionic chromatography (IC, Thermo Fisher ICS 3000) equipped with AS/AG 11HC and CS/CG 12A columns for anion and cation analyses, respectively. Methanesulfonate ($MeSO_3^-$) and carboxylic acids were also quantified in the same anionic run (Jaffrezo et al., 1998).

Anhydrosugar, sugar alcohol, and primary saccharide analyses were achieved using an HPLC with Pulsed Amperometric

Detection. A first set of equipment was used until March 2016, consisting of a Dionex DX500 equipped with three columns Metrosep (Carb 1-Guard + A Supp 15-150 + Carb 1-150), the analytical run being isocratic with 70 mM sodium hydroxide eluent, followed by a gradient cleaning step with a 120 mM NaOH eluent. This analytical technique enables detection of anhydrous saccharides (levoglucosan and its stereoisomers mannosan and galactosan), polyols (arabitol, sorbitol, mannitol), and glucose. A second set of equipment was used after this date, with a Thermo-Fisher ICS 5000+ HPLC equipped with 4 mm

diameter Metrosep Carb 2 × 150 mm column and 50 mm pre-column. The analytical run is isocratic with 15% of an eluent of sodium hydroxide (200 mM) and sodium acetate (4 mM) and 85 % water, at 1 mL min$^{-1}$ (Samake et al., 2018). The blank value was subtracted from the concentrations for both ions and sugars. Detection and quantification limits are presented in Table S1 of the supplementary material.

Further chemical species were determined with IC or HPLC-PAD. However, acetate, pyruvate, $NO_2^-$ and sorbitol were

excluded from the database due to experimental uncertainty and possible artifacts. From 2015 onwards, aerosol concentrations of inositol, glycerol, erythritol, xylitol, threalose and rhamnose were also obtained, but results are not included here due to the incompleteness of the data series and because for sorbitol, xylitol and rhamnose, most of the values were below the quantification limit. Formate, $F^-$, $Cl^-$ and $NO_3^-$ of $PM_{10}$-C were excluded from the analysis because, for this particular batch, the filters were analyzed three years after the sampling took place and acidic losses (Witz et al., 1990; Ashbaugh and Eldred,

2004) seem to have occurred.

**2.5 Statistical analysis**

Concentration measurements are not distributed normally and there are some outliers for individual species in certain samples. Therefore, we used non-parametric statistical methods in order to describe the data sets. The strength of the relationship between species concentrations was evaluated with the Kendall rank correlation method ($\tau$) instead of the typical Pearson's

coefficient to avoid outliers pulling the relationship. Comparison of average values between two datasets was made with



Wilcoxon rank sum test (Table S4). Statistical significance was set at $p < 0.05$ unless otherwise indicated. The median is a better statistic to describe our data, hence the concentrations are reported as the median ($\tilde{x}$) ± standard deviation ($\sigma$) in the manuscript. However, most of the literature reports mean average concentration, and therefore we will use it in the comparative figures. Mean and median concentrations and standard deviation for all species can be found in the supplementary material (Tables S2 and S3).

In order to find a structure in the seasonality patterns, the species were grouped by two clustering methods: k-means and hierarchical clustering (Govender and Sivakumar, 2020; Abdalla, 2022). For both cases, the variable to be analyzed was the monthly median concentration based on the Euclidean distance. The hierarchical clustering was made using the Ward method (Govender and Sivakumar, 2020 and references therein).

**2.6 Air-mass origin**

HYSPLIT (Hybrid Single Lagrangian Integrate Trajectory, Stein et al., 2015) back-trajectories are used to assess the origin of the air-masses for each filter. Hysplit desktop version 4.8 was used to obtain 96-hour back trajectories (without taking into account regional precipitation), run every hour. Only the periods when the high-volume sampler was working were selected. The working time of each filter was corrected if there was a reason for a change in the pre-defined schedule (power outage, mistake in programming the timer, etc.). This produced on average 67±38 air mass back trajectories per filter for $PM_{2.5}$ and 94±47 for $PM_{10}$. The back trajectories corresponding to each sample were then plotted on a polar grid centered on Chacaltaya. We then counted for each cell of the polar grid, the number of hours that each trajectory passed over it, producing a higher density of points for the most common pathways followed by the airmasses (see more details of this representation in Koenig et al., 2021).

Two different meteorological inputs were used to drive the backtrajectory analysis:

1)      High resolution dataset from January 2012 to September 2016: based on WRF model nested in three domains (Figures S5, S6 and Table S5) down to 1 km resolution, with ERA-interim data as boundary conditions. Nine starting points at 500 m a.g.l. were used in a 3x3 grid over Chacaltaya, with the station coordinates as the central point. This dataset has a topography better represented than the coarse resolution dataset described below, and represents well the transport over the complex terrain of the region (e.g., uplift of Amazonian air masses through valleys). It was used in previous works (Chauvigné et al., 2019; Koenig et al., 2021) and we use it for both individual sample-to-sample and grouped analysis. For the 143 samples covered by this dataset, only 15% of the samples had air masses being brought from a single dominant direction (W, NW, N, NE), 29% for two dominant directions (W, NW, N, NE, SW, SSW, S) and 49% of very mixed origins (no information for 7% of the samples).

2)      Coarse resolution dataset from January 2012 to April 2021: based on data from ERA-5, which has 30 km resolution and a more smoothed topography. This dataset cover the entire sampling period. The starting point of the backtrajectory was set at 1500 m a.g.l (of the ERA-5 defined topography) which corresponds to 528 hPa in the pixel containing the station



coordinates. Selection of this initial pixel was done to overcome the influence of the coarse topography of the reanalysis data.

This choice, however, produces a long-range biased backtrajectory due to the higher wind speed at upper levels. This dataset covers all samples, but sample to sample comparison with the nested-WRF backtrajectories shows only 43% of coincidences in transport patterns, although for long-range grouped back trajectories, there is overall agreement (Figure S8). We thus use this dataset for the interpretation of the entire record, using grouped samples (section 3.5).

## 3 Results and discussion

Mass concentrations were transformed to standard cubic meters (1013hPa, 0°C, see supplementary material) for all the sites presented in Table 4. For Chacaltaya, ambient concentrations can be found in Table S2. Concentrations are reported by pooling day and night samples (Table 2) because day and nighttime concentrations were statistically similar for the majority of the species (Table 5, section 3.5). Although $PM_{2.5}$ and $PM_{10}$ measurements were not simultaneous, we have compared the average mass concentrations for major species considering that the two fractions were sampled during long periods (>1yr). The species

concentrations in three $PM_{10}$ batches and the $PM_{2.5}$ batch were statistically similar for most of the species (Table S4), particularly in the wet season (DJFM). This hints to a predominant concentration of most species in $PM_{2.5}$ fraction, long-range transport of small particles, and in some cases co-emitting or co-located sources. For the three $PM_{10}$ groups, there was variability from group to group and season to season (Table S4), indicating highly variable environmental controls.

**Table 2. Bulk mass concentration (ng m$^{-3}$) of EC, OC, and species measured in Mt. Chacaltaya aerosol. Mean ($\bar{x}$), median ($\tilde{x}$), standard deviation ($\sigma$) and number of samples above the quantification limit (N) reported. Volume taken to standard cubic meters (1013hPa, 0°C). Data segregated by seasons and also at ambient conditions can be found in Tables S2 and S3.**

| Species | $PM_{10}$-A | | | | $PM_{2.5}$ | | | | $PM_{10}$-B | | | | $PM_{10}$-C | | | |
|---|---|---|---|---|---|---|---|---|---|---|---|---|---|---|---|---|
| | December 2011 to July 2013 | | | | August 2013 to Dec. 2015 | | | | April 2016 to August 2017 | | | | February 2019 to March 2020 | | | |
| ng m$^{-3}$ | $\bar{x}$ | $\tilde{x}$ | $\sigma$ | N | $\bar{x}$ | $\tilde{x}$ | $\sigma$ | N | $\bar{x}$ | $\tilde{x}$ | $\sigma$ | N | $\bar{x}$ | $\tilde{x}$ | $\sigma$ | N |
| OC | 913 | **703** | (648) | 56 | 881 | **669** | (729) | 77 | 801 | **605** | (604) | 49 | 607 | **529** | (449) | 32 |
| EC | 92.8 | **68.2** | (90.8) | 36 | 80.7 | **69.7** | (48.6) | 74 | 77.7 | **56.5** | (68.7) | 49 | 65.8 | **51.3** | (48.5) | 32 |
| Li$^+$ | 0.05 | **0.04** | (0.07) | 48 | 0.02 | **0.03** | (0.03) | 52 | 0.012 | **0.005** | (0.023) | 19 | 0.015 | **0.013** | (0.012) | 27 |
| Na$^+$ | 25.2 | **21.9** | (20.8) | 54 | 25.2 | **20.1** | (15.4) | 71 | 29.9 | **22.7** | (26.8) | 41 | 12.05 | **9.17** | (10.3) | 31 |
| NH$_4^+$ | 158 | **145** | (106) | 58 | 305 | **295** | (192) | 78 | 316 | **306** | (196) | 49 | 281 | **248** | (161) | 32 |
| K$^+$ | 24.2 | **17.7** | (22.8) | 54 | 26.4 | **23.3** | (17.8) | 77 | 34.1 | **15.5** | (34.0) | 47 | 25.7 | **19.9** | (26.3) | 31 |
| Mg$^{+2}$ | 7.28 | **6.83** | (4.51) | 56 | 8.62 | **7.87** | (5.51) | 75 | 6.86 | **5.04** | (6.66) | 49 | 6.66 | **5.76** | (4.89) | 32 |
| Ca$^{+2}$ | 72.4 | **66.4** | (43.0) | 55 | 90.8 | **74.5** | (78.4) | 79 | 57.4 | **51.3** | (43.0) | 48 | 51.4 | **36.2** | (42.0) | 32 |
| F$^-$ | 2.05 | **1.48** | (2.31) | 49 | 2.80 | **1.79** | (2.49) | 76 | 1.49 | **1.13** | (1.17) | 42 | | *Lost* | | 14 |
| Cl$^-$ | 8.72 | **6.17** | (7.08) | 47 | 13.3 | **9.79** | (11.0) | 45 | 10.85 | **7.73** | (11.91) | 34 | | *Lost* | | 25 |
| Br$^-$ | 1.45 | **1.27** | (0.92) | 50 | 1.18 | **0.91** | (0.79) | 69 | 1.63 | **1.09** | (1.54) | 38 | 0.96 | **0.78** | (0.80) | 28 |
| NO$_3^-$ | 149 | **128** | (114.2) | 58 | 84.9 | **65.6** | (79.2) | 76 | 110 | **71.1** | (115.8) | 45 | | *Lost* | | 29 |
| SO$_4^{-2}$ | 451 | **399** | (327) | 58 | 1121 | **1047** | (795) | 79 | 991 | **920** | (761) | 52 | 905 | **889** | (499) | 32 |
| MeSO$_3^-$ | 6.58 | **6.67** | (3.56) | 54 | 7.86 | **7.45** | (4.41) | 76 | 6.73 | **6.22** | (3.45) | 48 | 4.52 | **4.83** | (2.48) | 32 |
| HCO$_2^-$ | 12.3 | **8.78** | (10.7) | 48 | 13.8 | **12.0** | (6.99) | 50 | 27.6 | **9.80** | (68.9) | 16 | | *Lost* | | 27 |
| C$_2$O$_4^{-2}$ | 32.8 | **22.6** | (40.8) | 57 | 44.0 | **40.3** | (30.5) | 76 | 38.0 | **30.3** | (37.4) | 45 | 53.2 | **43.6** | (31.7) | 32 |
| Glucose | 3.29 | **2.64** | (2.10) | 54 | 3.69 | **3.04** | (2.64) | 48 | 2.91 | **2.46** | (1.75) | 44 | 1.81 | **1.59** | (1.12) | 23 |
| Levoglucosan | 12.6 | **5.3** | (22.7) | 56 | 14.3 | **8.47** | (18.9) | 70 | 36.6 | **26.5** | (41.1) | 32 | 11.0 | **6.77** | (10.6) | 19 |
| Mannosan | 2.92 | **0.99** | (5.5) | 17 | 2.06 | **0.80** | (3.7) | 50 | 6.93 | **4.07** | (9.51) | 25 | 1.38 | **1.10** | (1.32) | 14 |
| Galactosan | 1.99 | **0.76** | (4.92) | 23 | 1.38 | **0.57** | (2.26) | 48 | 5.01 | **2.35** | (8.09) | 26 | 1.09 | **1.04** | (0.94) | 14 |
| Arabitol | 3.76 | **2.63** | (4.31) | 58 | 1.67 | **1.45** | (0.88) | 44 | 1.04 | **0.96** | (0.32) | 31 | 0.56 | **0.51** | (0.38) | 22 |
| Mannitol | 1.38 | **1.09** | (1.10) | 49 | 1.69 | **1.37** | (1.24) | 43 | 1.85 | **1.63** | (1.04) | 40 | 1.17 | **0.92** | (0.87) | 13 |



## 3.1 Mass and mass closure

Aerosol mass for $PM_{10}$ is presented on Table 3, and it ranges from 2.4 to 22.6 µg m$^{-3}$ (at ambient conditions: 1.1-12.0 µg m$^{-3}$).
Even though this dataset is limited in temporal extent, it is consistent with measurements made on mountain sites, like Pûy de Dôme in France (5.6±4.6 µg m$^{-3}$, Bourcier et. al., 2012), Mt. Atlas in Morocco (11.8-14.5 µg m$^{-3}$, Deabji et al., 2021), Mt. Everest in Nepal (10 µg m$^{-3}$, Decesari et al., 2010); Montsec in Spain (2.9-20.6 µg m$^{-3}$; Ripoll et al., 2015), and Izaña at the Canary Islands (10.9-50.5 µg m$^{-3}$; García et al., 2017a, b). The highest measured mass coincides with a dry month and the start of the eruptions of Sabancaya volcano in November 2016 (Moussallam et al., 2017).

For mass closure estimation, organic matter (OM) was calculated based on OC concentrations. As source contributions vary with the seasons, it is not possible to assign a single conversion factor between OC and OM. Therefore, it was estimated for two cases : a minimum value using a factor of 1.8 as for other mountain sites and a maximum value using a factor of 2.6 as for aged aerosol (Cozic et al., 2008; Chow et al., 2015 and references therein), so organic contributions were estimated to range from 7 to 34% for the available series.

The contribution of mineral dust is very important, especially for the dry season, accounting for 33-56% of $PM_{10}$. This fraction is higher than in La Paz – El Alto, where mineral dust contributes to 20-35% of the mass (Mardoñez et al., 2022). Secondary inorganic aerosol (SIA= $NH_4^+ + NO_3^- + SO_4^{-2}$) ranged from 15 to 26%, with all sulphur found in soluble form. The unaccounted fraction (ND) is related to heteroatoms and water content, and it usually ranges from 20 to 40% (Tsyro, 2005; Hueglin et al., 2005). For our case, the non-determined fraction ranged from 2 to 35%, being lower particularly in the wet season. This may
be due to wash out of hygroscopic species and maybe also to less heteroatoms present in the organic fraction, though this is still under study.

**Table 3. Chacaltaya STP concentrations of $PM_{10}$ total mass, mineral matter (MM), secondary inorganic aerosol (SIA), elemental carbon (EC), the sum of trace and rare elements (TE), organic matter minimum and maximum estimations, and the fraction of non-**
**determined (ND) aerosol, its range defined by the OM variation. No ion data for B49 and B50. For ambient conditions, divide concentrations by 1.89**

| Sample | Start and end date | $PM_{10}$ µg m$^{-3}$ | MM µg m$^{-3}$ | SIA µg m$^{-3}$ | EC µg m$^{-3}$ | TE ng m$^{-3}$ | OM µg m$^{-3}$ | ND % | Measured $PM_{10}$/Σions |
|---|---|---|---|---|---|---|---|---|---|
| B42 | 04/11 – 11/11/2016 | 22.58 | 7.43 | 3.41 | 0.16 | 116 | 3.56-5.14 | 35-28% | 5.7 |
| B44 | 30/12/2016 – 06/01/2017 | 2.78 | 0.98 | 0.42 | 0.04 | 15.2 | 0.66-0.96 | 24-13% | 6.3 |
| B45 | 03/02 – 10/02/2017 | 8.04 | 3.60 | 1.98 | 0.08 | 47.2 | 1,48-2.13 | 11-2% | 3.7 |
| B46 | 03/03 – 10/03/2017 | 2.44 | 1.02 | 0.63 | 0.02 | 16.0 | 0.41-0.59 | 14-7% | 3.8 |
| B48 | 12/05 – 19/05/2017 | 15.12 | 8.41 | 2.29 | 0.06 | 109 | 1.09-1.57 | 21-18% | 6.1 |
| B49 | 23/06 – 30/06/2017 | 12.44 | 6.97 | - | - | 89.2 | - | - | - |
| B50 | 04/08 – 11/08/2017 | 21.87 | 10.34 | - | - | 140 | - | - | - |





### 3.2 General composition of particulate matter and comparison to other sites

#### 3.2.1 Dominant ions and carbonaceous species

Mass concentrations determined at Chacaltaya for ions, total carbon (EC+OC) and saccharides present an interquartile range
of 917 to 2308, 519 to 1175, and 3.9 to 24.1 ng m$^{-3}$ for PM$_{2.5}$ respectively; and 558 to 1785, 384 to 1120, and 4.3 to 25.5 ng
m$^{-3}$ for bulk PM$_{10}$. Chacaltaya PM$_{10}$ is compared to other available high-altitude sites (Northern Hemisphere) and other selected
sites in South America listed on Table 4 and presented in Figure 2.

**Table 4. Sites used in the comparative Figures 2 and 3. The ABL-TopoIndex classifies high-altitude sites according to the influence
of the atmospheric boundary layer on them (the lowest the value, the less influenced by the convective planetary boundary layer,
Collaud Coen et al., 2018), which is available for some of the sites.**

| | Acronym and location | | Lat. | Lon. | Alt. m.a.s.l | ABL Topo Index | Type | References |
|---|---|---|---|---|---|---|---|---|
| High-altitude | CHC | Chacaltaya, Andes, western Bolivia | -16.3470 | -68.1283 | 5380 | 1.34 | High altitude, regional background | This work |
| | NCO | Nepal Climate Observatory, Mt. Everest, Nepal | 27.9578 | 86.8149 | 5079 | 3.43 | High altitude, regional background | Decesari et al., 2010 |
| | IZO | Izaña, Canary Islands, Spain | 28.3090 | 16.4994 | 2373 | 0.57 | High altitude, island mountain top, background | García et al., 2017a,b |
| | RUN | Maïdo Observatory, Reunion Island | -21.0795 | 55.3831 | 2160 | 0.79 | High altitude, island mountain top, background | Dominutti et al., 2022 |
| | AMV | Atlas Mohamed V Observatory, Middle Atla,s, Morocco | 33.4062 | -5.1033 | 2100 | - | High altitude, regional background | Deabji et al., 2021 |
| | MSC | Montsec, Spain | 42.0500 | 0.7333 | 1570 | 2.07 | High altitude, continental background | Ripoll et al., 2015 |
| | PDD | Pûy de Dôme, Massif Centrale, France | 45.7723 | 2.9658 | 1465 | 2.72 | High altitude, continental background | Bourcier et al., 2012 |
| | EA | El Alto city, Bolivia | -16.5100 | -68.1987 | 4025 | - | Urban near Chacaltaya, high altitude but not mountain sites | Mardoñez et al., 2022 |
| | LP | La Paz city, Bolivia | -16.5013 | -68.1259 | 3600 | - | | |
| Forest and rural | RBJ | Jaru natural reserve, Rondônia state, Brazil | -10.0819 | -61.9300 | 110 | | Forest, background | Graham, 2002 |
| | ZF2 | Cuieiras ZF2 natural reserve (TT34), Amazonas state, Brazil (+54m tower) | -2.6091 | -60.2092 | 110 | | Forest, background | Custodio et al., 2019 |
| | ATTO | Amazonian Tall Tower Observatory (+80m tower), Amazonas state, Brazil | -2.1465 | -59.0218 | 120 | Not applicable | Forest, background | Barbosa, 2014 |
| | BAL-01 | Balbina, Amazonas state, Brazil | -1.9166 | -59.4000 | 174 | | Forest, background | Claeys et al., 2004 Elbert et al., 2007 |
| | FNS | Fazenda Nossa Señora Aparecida, Rondônia state, Brazil | -10.7622 | -62.3575 | 315 | | Rural (biomass burning campaign) | Kundu et al., 2010a |
| | SHU | Shuara 9, Sucumbíos, Ecuador | -0.0592 | -76.5603 | 250 | | Rural background | Barraza et al., 2020 |
| | | Auca Sur, Orellana, Ecuador | -0.7046 | -76.8878 | 277 | | Rural background | Barraza et al., 2020 |
| | FLO | La Florida, Esmeraldas, Ecuador | -0.9317 | -79.6780 | 72 | | Rural background | Barraza et al., 2020 |

Ions and EC-OC concentrations at Chacaltaya station are lower than other high-altitudes sites, with the exception of EC-OC
for the sub-tropical station of Izaña and the tropical station of Maïdo, for which concentration levels are similar to Chacaltaya.
For Izaña, nitrate and stands out as a long-range pollutant transported to the observatory (Rodríguez et al., 2011) while for
Chacaltaya its concentration is proportionally lower. Mt. Atlas presents a higher load of anions and cations relative to Mt.
Chacaltaya, mostly related to important dust and urban influences at a broad regional scale. Pûy de Dôme, Montsec, and Mt.
Everest present a higher influence of pollution (SO$_4^{-2}$, NO$_3^-$, EC, OC) compared to Mt. Chacaltaya, because the former lie in
the populated western Europe and are influenced by regional pollution (Bourcier et al., 2012; Ripoll et al., 2015; Querol et al.,
2013), and because the latter is influenced by long-range north Asiatic pollution in spite of its remote location (Bonasoni et



al., 2010; Decesari et al., 2010). EC, OC and chloride concentrations at Chacaltaya can be more than five times lower than urban sites in the region, and Cl⁻ is also lower than sites influenced by relatively fresh and unchanged marine aerosol (such as ATTO, Barbosa 2014). Combustion tracers (OC, EC, K⁺, section 3.3) at Chacaltaya are lower compared to sites showed in Figure 2. Chacaltaya concentrations are similar to the tropical broadleaf forest site of Cuieras ZF2, with the exception of a
significantly higher OC, K⁺, Na⁺ contribution in ZF2 due to the important influence of both biomass burning and biogenic emissions at the forest site.

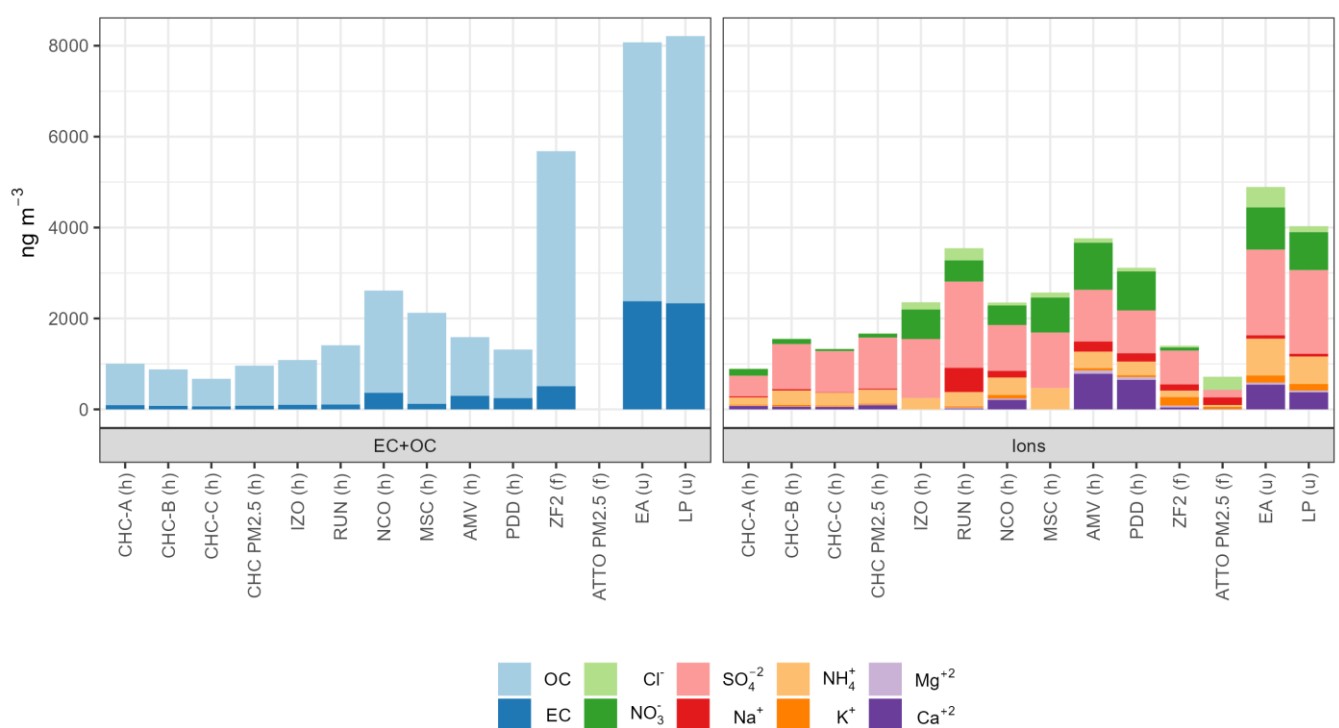

**Figure 2. Aerosol chemical composition at standard conditions of several sites compared to Chacaltaya. PM$_{10}$ is reported for all the**
**sites, except for ATTO which has only PM$_{2.5}$ information. Note that Na⁺, K⁺, Mg⁺², Ca⁺² are not available for IZO, MSC; and EC, OC for ATTO.**

For Chacaltaya, OC and SO$_4^{2-}$ are the two dominant components identified, both adding up to 70% of the measured mass of each size fraction, followed by NH$_4^+$ (which neutralizes sulfate) and NO$_3^-$, which together contribute 17% to the aerosol mass. The dominance of OC and sulfate is similar to most of the other sites shown in Figure 2, but sources are different. For instance,
for the Cuieras forest site, OC represents alone 70% due natural and biomass burning influences. At Izaña, sulfate is mainly related to long-range transport of anthropogenic emissions (Rodríguez et al., 2011; García et al., 2017a) and at Mt. Everest, OC and sulfate emissions are heavily influenced by coal burning (Bonasoni et al., 2010; Decesari et al., 2010). At Chacaltaya, sulfate can have a variety of sources: sea salt, soil dust (Surkyn et al., 1983), fossil fuel combustion (diel cycle observed in Chacaltaya for PM$_1$, Bianchi et al., 2021), marine secondary aerosol (Sholtz et al., 2023), and volcanism (Aliaga et al., 2021).





Impact of coal combustion in Chacaltaya is discarded given that coal is not used in the tropical Andes (EANET, 2019) as corroborated by the absence of lignite markers in the samples (Figure S4). Sea-salt sulfate contributions account only for 3% for $PM_{10}$-A and 1% for $PM_{10}$-B, C and $PM_{2.5}$ (calculated as $[seaSO_4^{2-}]=2.06 [Mg^{2+}]$ ng m$^{-3}$ based on Keene (1986) ratios with $Mg^{2+}$ chosen because of fewer missing values than for $Na^+$). Nitrate and OC are notably related to regional biomass burning (section 3.3) and urban emissions from the metropolitan area nearby (Bianchi et al., 2021), though in the wet season, part of

the organic fraction has been related to natural sources (Zha et al., 2023). $NH_4^+$ is discussed in section 3.5. Although Chacaltaya ion concentrations are lower than northern hemisphere stations, this Andean site doubles the total soluble fraction ($\Sigma_{ions}$=1.7 µg m$^{-3}$ in $PM_{2.5}$) measured at the pristine Amazonian Tall Tower Observatory ($\Sigma_{ions}$=0.7 µg m$^{-3}$ in $PM_{2.5}$ from Barbosa, 2014). In summary, the Chacaltaya GAW station is a regional background site with relatively low ion and EC-OC concentrations when compared to other high-altitude stations in the world and low-altitude background stations in the region (with the notable

exception of the Amazonian Tall Tower Observatory).

### 3.2.2    Organic and elemental carbon

Elemental carbon (EC) is related to fine particles directly emitted by combustion processes (typically traffic and biomass burning) and organic carbon (OC) to both primary and secondary organic aerosol (SOA). The OC/EC ratio is used as a proxy for the nature of carboneous primary sources (Salma et al., 2004) but also of the aging of aerosols (Waked et al., 2014). The

OC/EC mass ratio in Chacaltaya was 10.9±4.5 for bulk $PM_{10}$ and 11.0±7.2 for $PM_{2.5}$, similar to sites classified as rural (8-11 Querol et al., 2013). This mass ratio is much higher than the observed for urban areas in South America (e.g. El Alto 2.6±1.1 from Mardoñez et. al 2022; Jaciara 2.0 ±1.2, Campo Novo dos Parecis 3.0±1.9, from Custodio et al., 2019) and closer to the natural environment of Cuieiras forest (12.2±2.2 for $PM_{10}$ and 6.6±1.0 for $PM_{2.5}$ Custodio et al., 2019). It is lower than for remote locations where aerosol arrives quite aged (13 in Siberia, Mikhailov et al., 2017; 12 in remote China, Zhang et al.,

2008; 12-15 for Izaña and Montsec, Querol et al., 2013) and forest burning emissions near the source (14.5 Watson and Chow, 2001).

In Chacaltaya, the OC/EC ratio is quite similar over the year: median value is 10.5±7.0 for DJFM, 10.4±6.0 for MJJA, and 10.9±4.5 for SON. Not even the outstanding biomass burning season (JAS) produces a ratio (10.4±4.4) different from the rest

of the year. This is contrasting with what has been observed in other sites such as Mt. Atlas, where OC/EC ratio showed marked seasonality (summer: 11.2, winter: 2.2, Deabji et al., 2021). The measured ratios and their uniformity over the year seem to be influenced by two factors: First, air-masses that are relatively aged during long-range transport, and second, the quick transformation (in ca. two hours) of short-range urban emissions such as observed over other urban areas (DeCarlo et al., 2008). We hypothesize that the high UV of the tropical atmosphere over the Altiplano could play a role in the impressively

fast aging of the organic matter at this site when transported from the nearby urban area.





Secondary organic aerosol was estimated for bulk $PM_{10}$ using the Castro et al. (1999) approximation ($OC_{secondary} = OC_{total} - $ [levoglucosan]*[OC/levoglucosan]$_{min}$ − EC*[OC/EC]$_{min}$) and it was found that SOA contributes approximately 50% of the OC mass during the dry season and 69% during the wet season for the entire dataseries. During the wet season, other organic
markers such as formate, oxalate, arabitol, mannitol and glucose are present, suggesting that important SOA formation processes are not only due to anthropogenic but also to natural precursors.

### 3.2.3    Saccharides

In Figure 3 we present a comparison of arabitol, mannitol and glucose, which are tracers of primary biogenic aerosol, at several sites. Glucose is the most abundant sugar in vascular plants (Opsahl and Benner, 1999) and consequently, it is found in plant
fragments (Pietrogrande et al., 2014; Graham et al., 2003), suspended soil from cultivated land (Rogge et al., 2007), biomass burning products and pollen (Elbert et al., 2007; Kundu et al., 2010a; Claeys et al., 2010). Indeed, pollen is transported to the Chacaltaya region mostly from Puna herbaceous vegetation (Altiplano) and Yungas (Subandes) trees, with a marginal evergreen Amazonian contribution (Brugger et al., 2019).  Arabitol and mannitol are emitted along with fungal (Bauer et al., 2008) and fern spores (Graham et al., 2003), but also found in leaves, pollens and green algal lichens (Medeiros et al., 2006;
Jia et al., 2010). In Balbina, Amazonia, mannitol has been linked to nocturnal Basydiomycota emissions (Elbert et al., 2007).



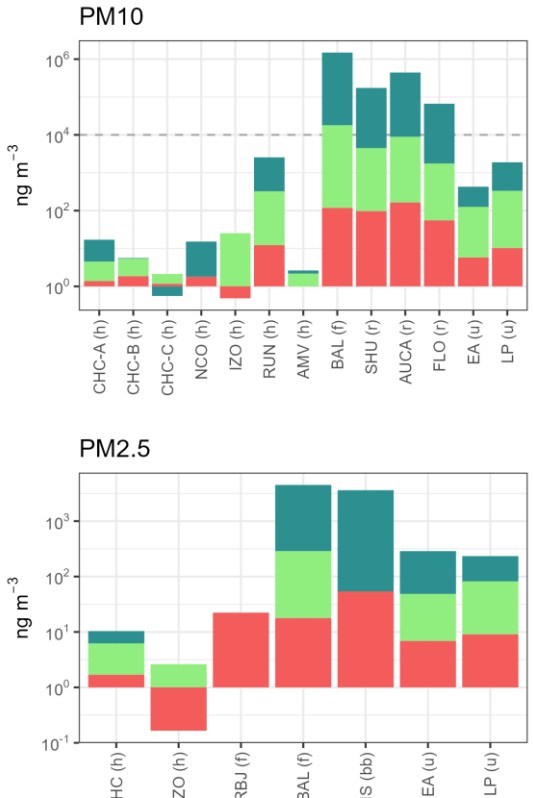

**Figure 3. Comparison of biogenic markers (mean concentrations) among different sites taken to standard cubic meters for PM$_{10}$ (upper panel) and PM$_{2.5}$ (lower panel). Mind the logarithmic scale. For AMV, only cases without Saharan dust influences were included. IZO, RUN, FNS data are for short term campaigns. When a species is not plotted, it indicates that it was not measured at the site. Dashed line in PM$_{10}$ shows upper limit of PM$_{2.5}$ plot.**

In general, polyols tend to be more concentrated in PM$_{10}$ than in PM$_{2.5}$ (Yttri et al., 2007; Samake et al., 2018; Elbert et al., 2007; Brighty et al., 2021; Zhang et al., 2015), especially in dry conditions (Rathnayake et al., 2017), but they are shifted to PM$_{2.5}$ when the cells rupture under moist conditions (Rathnayake et al., 2017; Yttri et al., 2007) and therefore they are not very affected by rainout. This difference between PM$_{10}$ and PM$_{2.5}$ is not pronounced at Chacaltaya, likely indicating fragmented biogenic material that is present mostly in PM$_{2.5}$ and/or loss of coarse material during long range transport.

Chacaltaya concentrations are highly variable, as can be seen in PM$_{10}$-A, B and C barplots in Figure 3. Saccharide data for high-altitude sites are scarce, but for what are available, Chacaltaya bulk PM$_{10}$ concentrations (glucose 2.9±1.9 ng m$^{-3}$, arabitol 1.4±3.4 ng m$^{-3}$, mannitol 1.5±1.1 ng m$^{-3}$) are similar to other semi-arid environments such as Mt. Everest and Mt. Atlas, but lower than Maïdo observatory in the tropical Southern Ocean, and rural sites either from tropical (Brazil, Ecuador; Figure 3) and extra tropical regions (e.g. France form Samake et al., 2018). This difference is more pronounced for PM$_{10}$, depicting an



important influence of a region where vegetation is scarce (Altiplano), in agreement with other studies (Brugger et. al., 2019). This is interesting, because even if the highly productive Amazonian forest is in the windward side (Figure 1) of the observatory it does not to have a big influence at our site in terms of saccharides, probably due to dilution, loss of coarse primary biogenic aerosol in the transport and/or unstudied degradation mechanisms of sugars along the way.

At Chacaltaya, mannitol concentrations are similar for $PM_{10}$ and $PM_{2.5}$, while for arabitol this is not so clear (Table S4). Their correlation is poor to moderate ($\tau$=0.54, 0.29, 0.47, 0.54 for $PM_{2.5}$, $PM_{10}$-A,B,C, respectively), contrary to what was found for a significant number of sites (Marynowski and Simoneit, 2022 and references therein), indicating emissions of endemic fungal species. Maximum values of arabitol were found in the $PM_{10}$-A fraction, but later on (Figure S3) concentrations show a significant decrease. Glucose correlates moderately well with mannitol except for $PM_{10}$-C ($\tau = 0.67$, 0.51, 0.67, and 0.27)

where it correlates with arabitol ($\tau = 0.36$, 0.30, 0.43, 0.67), the explanation for which remains unclear.

### 3.3 Observed seasonality of the atmospheric species

Long-term series (Figure S3) show clear seasonality for most of the species, and seasonal patterns were obtained from monthly boxplots. The number of samples available per month are presented in Figure 4, which shows that at least 10 samples per month were available. In Figure 5, monthly boxplots for each species are presented. The boxplots were made by aggregating

all the sampled periods. Seasonal patterns can be classified in three principal groups based on clustering of their normalized median monthly concentrations (Figure S11). These groups are: 1. Species with high concentration during the dry months, 2. Species with maxima concentration between July and September, 3. Species with unclear seasonality patterns. However, there are nuances within this classification and they will be discussed in the text.

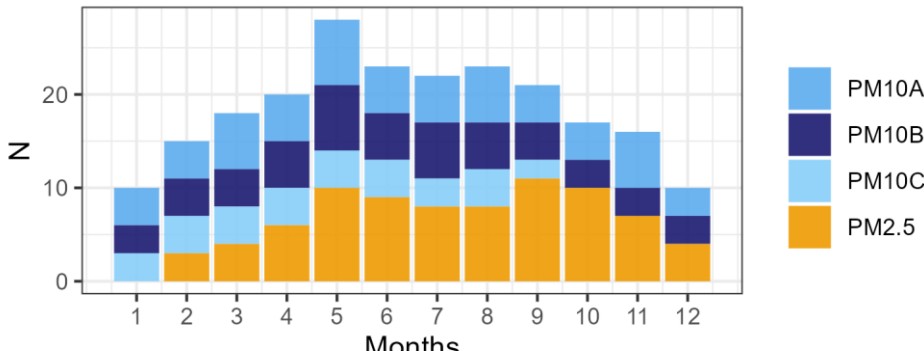

**Figure 4. The number of valid samples per month used in this study. The color represents the sampling period (see Table 1 for details).**

### 3.3.1 Species with high concentration during the dry months

The ions $MeSO_3^-$, $SO_4^{2-}$, $NH_4^+$, $Na^+$, $Mg^{2+}$, and $Ca^{2+}$ present a seasonality opposed to the seasonality of moisture (Figure S13) and related to the seasonal increase of westerlies at the station (Scholz et al., 2023), having the lowest values in the wet-season

(DJFM), and higher values during the dry months of the year, with an abrupt transition between November and December



(Figure 4). In the wet-season, air masses arrive mostly from Amazonia (N, NE, Figure S7.), typically along with precipitation, and these species can be affected by rainout or simply not be emitted (such as mineral dust from wet soils). For the rest of the year, air resides more over the Altiplano, arriving to the station from three main directions as identified by Chauvigné et al (2019) and shown in Figure S7: east-south east (Amazonia and Chaco), northwest (Manu, Madidi, national parcs) and west (eastern Bolivia, southern Peru). The further away the air mass is coming, the lower the influence from the surface (Aliaga et al., 2021). All these species arrive together to the station due to the large-scale atmospheric circulation seasonality, but have different sources.

Sulfate and ammonia correlate well ($\tau$ =0.81), likely because of the formation of $HNH_4SO_4$ and $(NH_4)_2SO_4$ that can be transported over long distances. Sulfate sources were mentioned in section 3.2.1 and ammonia sources are discussed in section 3.5. Methanesulfonate is a well-known tracer of phytoplankton emissions (Finlayson-Pitts and Pitts Jr, 1999), as biogenic dimethyl sulfide is transformed into $MeSO_3^-$ in the atmosphere. Median $MeSO_3^-$ concentrations in Chacaltaya (bulk $PM_{10}$: 6.03±3.40, $PM_{2.5}$ 7.45±4.41 ng m$^{-3}$) are twice the concentration at La Paz-El Alto. It has been shown that $MeSO_3^-$ is transported to Chacaltaya from the Pacific Ocean (at least 330 km from the station but as far as >1600 km) through the low free troposphere with a negligible contribution from Titicaca Lake (Scholz et al., 2023).

The ions $Ca^{2+}$, $Mg^{2+}$, and $Na^+$ seem to have a dominant continental origin. Non-sea-salt $Ca^{2+}$ accounts for 97% of $Ca^{2+}$. Median $Ca^{2+}/ Mg^{2+}$ mass ratio is 8.8±4.1 and 10.6±4.7 for bulk $PM_{10}$ and $PM_{2.5}$, respectively, with slightly higher values in the dry season (12.1±3.6 and 11.2±3.6), in agreement with very dry soils and strong winds typical of that time of the year. Indeed $Ca^{2+}/ Mg^{2+}$ mass ratio is closer to continental stations values ($PM_{10}$: Mt. Atlas 10.8, El Alto 10.3, $PM_{2.5}$: Arequipa 8.8, El Alto 9.5) than to sea salt (0.31 Keene, William C. et al., 1986). For $PM_{10}$-B, the median soluble fraction for $Ca^{2+}$, $Mg^{2+}$, and $Na^+$ was 30±25%, 10±8% and 10±18%, respectively, indicating an important crustal influence. However, a marine contribution seems also possible especially for a fraction of $Mg^{2+}$ and $Na^+$, hinted by their clustering together (Figure S11), and in agreement with a marine source affecting this site as hypothesized by Adams et al. (1977). Finally, and in agreement with our findings, $SO_4^{-2}$, $Ca^{2+}$, $Mg^{2+}$, and $Na^+$ were identified as a group of species influencing the region from the long-range (Mardoñez et al., 2022).








**Figure 5. Monthly boxplots for Chacaltaya, all samples together (same color scale as Figure 4). F⁻, Cl⁻, formate and NO₃⁻ were not included for PM₁₀-C due to suspected losses. Boxplots were produced with at least five points, with top, middle and bottom lines for the 75th, median and 25th percentiles, respectively and the whiskers for the 5% and 95% confidence intervals. Extreme outliers are plotted as triangles, and their values are in the supplementary material. Species are ordered according to their seasonality pattern.**
**Shading highlights maximum input of biomass burning products in July, August and September (section 3.3.2.), and April to November (section 3.3.1). Plots with no shading are the group of species without a clear seasonal pattern (section 3.3.1)**





### 3.3.2 Species with maxima concentration between July and September

Levoglucosan, $K^+$, $NO_3^-$, EC, OC, formate, oxalate, $F^-$, $Cl^-$, and $Br^-$ constitute another group. They follow the same seasonality as levoglucosan[1]*, a proxy of biomass burning emissions, which shows concentrations above the annual average between June
and November, with notorious maxima concentrations between July and September. For this group, the transport conditions are essentially the same as in the previously described group. However, the main difference is that this group is heavily influenced by the seasonal activity of a dominant source, namely human-induced fires related to agriculture and deforestation. The regions active in terms of burning for June to November (within the range of four-day backtrajectories) are located in the Amazonian (Bolivia east of the Andes, and Brazilian states neighboring Bolivia, particularly those part of the "deforestation
arc") and Gran Chaco (southern Bolivia, northern Argentina and Paraguay) basins (Carmona-Moreno et al., 2005; Giglio et al., 2013; Dutra et al., 2022; Pereira et al., 2022). Southern Peru and Altiplano grassland fires have a very small influence compared to the aforementioned regions (Bradley and Millington, 2006).

Levoglucosan stereoisomer ratios can be used to assess the substrate that was burned (Marynowski & Simoneit 2022; Xu et al. 2019). For the entire dataseries, the interquartile range of the mass ratio levoglucosan/mannosan is 9.3-21.5 and
mannosan/galactosan is 0.94-1.67, depicting a wide variety of sources, with grass, duff/litter and hardwood being the main burning substrates, and a few events of peat fires (Figure S4). All this includes agricultural waste burning and also probably land clearing due to the hardwood fingerprint (forests being changed to pasture sites). Note however, that mannosan and galactosan are usually below the quantification limit in the wet months and therefore, ratios between December and April cannot be calculated even if levoglucosan is present. For the wet season the burning source is hypothesized to be nearby small-
scale fires (such as those observed in the close urban area, Mardoñez et al., 2022).

The transport mechanism of smoke to this site is complex. Burning emissions, along with other organic pollutants, are transported to the Andes (Magalhães et al., 2019; Estellano et al., 2008) through what would be the free troposphere from an Amazonian point of view (1-4 km a.s.l, Andreae et al., 1988; Kundu et al., 2010b), and a regional residual layer from the Altiplano point of view (> 5 km a.s.l., Figure S12). Additionally, smoke goes upslope in the valleys (Bourgeois et al., 2015),
as observed on higher influence of smoke in aerosol mass in La Paz valley (17% in mass) than at the Altiplano (13%, Mardoñez et al., 2022). Once the smoke arrives to the Altiplano, it is picked up by convection and it is transported with the PBL development to the station (along with urban pollutants, natural emissions, etc.). Hence, although the smoke primarily travels from east to west towards the Andes, it also reaches the mountains through a bottom-up flow originating from the western side of the Cordillera Real.

The species in this group can have other origins than solely biomass burning, as they are also observed throughout the rest of the year. Potassium seems to also have a biogenic influence. Its correlation with levoglucosan is good ($\tau$=0.63), but lower than near the source ($r^2$=0.8, Schkolnik et al., 2005; $r^2$=0.92 at pasture site, $r^2$=0.64 at forest site Graham, 2002), indicating the input

---

[1] Levoglucosan* = levoglucosan + mannosan + galactosan



of other sources of $K^+$ between the emission location and Chacaltaya, such as silico aluminates (clay minerals and feldespars), aged marine aerosols and/or primary forest emissions (Ascomycota spore discharge Elbert et al., 2007; Webster et al., 1995), the latter in agreement with other studies (Correia et al., 2003). Fluoride and bromide are not important constituents of vegetation, but plants can bio accumulate halogens in polluted environments (Jayarathne et al., 2014), so for the burning season the species may release agrochemicals to the atmosphere. Indeed, $F^-$ was observed in rainwater samples affected by agrochemicals (Zunckel et al., 2003), and in high concentrations >100 ng m$^{-3}$ near anthropized burning sources (Kundu et al., 2010a). Additionally, $F^-$ can have a volcanic source (Aiuppa, 2009; Aiuppa et al., 2009) that is active all year round, as it was detected in 81% the samples and more frequently in the 2014-2015 time period (yellow dots in Figure 4). For $Br^-$ a volcanic origin seems unlikely because of the very low values of HBr measured in emissions from Sabancaya volcano in 2015 (Moussallam, personnal communication). Instead, bromide seems to have a marine origin (along with Cl, I, Mg and Na; Adams et al., 1977), its concentrations in Chacaltaya (1.3±1.0 ng m$^{-3}$) being in the lower range of those reported in a coastal site (1 to 8 ng m$^{-3}$ in Cape Verde, Müller et al., 2010). Chloride can also have marine, volcanic and urban origins. The $Na^+/Cl^-$ ratio is 2.9±9.3, higher than marine ratio (0.56 Keene, 1986), but it is well established that $Cl^-$ can be lost to gaseous phase in the presence of atmospheric acids (Laskin et al., 2012 and references therein). A hint for a volcanic origin of chloride is found in some samples with high $Cl^-$ content coinciding with the onset of eruptive activity of Peruvian volcanoes (Table S6). The influence of an urban source (seen in Mardoñez et al., 2022 as litter burning) seems small, as $Cl^-$ was <1% in $PM_1$ mass (Bianchi et al 2021). Nitrate has permanent urban influence as a secondary inorganic aerosol (section 3.5), long-range transport to the region (Mardoñez et al., 2022) and probably some lightning (Wang et al., 2021) in the wet season. Finally, formate (12.0±7.0 ng m$^{-3}$ for $PM_{2.5}$, 6.0±30.4 ng m$^{-3}$ for bulk $PM_{10}$) and oxalate concentrations (40.3±30.5; 32.4±38.3 ng m$^{-3}$) are in the lower end of those measured in continental environments ($HCO_2^-$ 9-239 ng m$^{-3}$ Yu, 2000; $C_2O_4^{-2}$: 20-400 ng m$^{-3}$ Golly et al., 2019). Oxalate seems to have an urban origin (Kawamura and Kaplan, 1987) because it clusters with EC (Figure S11). However, other processes (such as biogenic emissions Kundu et al., 2010b; Yamasoe et al., 2000; oxidation of different carboxylic acids Yang et al., 2014, in-cloud formation of $C_2O_4^{-2}$ Zhang et al., 2017) are most probably involved.

### 3.3.3 Species with unclear seasonality patterns

No straight seasonal pattern can be inferred for $Li^+$, arabitol, mannitol, and glucose from Figure 4. Hence, for these species, it seems that there are other drivers of their climatology, as they do not fit the narrative of a clear emission source (such as the levoglucosan group) nor the circulation pattern change ($MeSO_3^-$ group).

Soluble lithium contributions (0.03±0.05 ng m$^{-3}$) are barely higher than the quantification limit, and present relative maxima between September and November, probably in relation to biomass burning. Additionally, the semi-arid environment where Chacaltaya is located, along with salt flats located south (ca. 400 km) can have an influence as elemental Li at this site. It was observed that backtrajectories dominated by Amazonian airmasses (N, NNE) present markedly different concentrations of elemental Li (0.04 to 0.05 ng m$^{-3}$) compared to the rest of the origins (0.18 to 0.38 ng m$^{-3}$).



Glucose, mannitol, and arabitol present high variability, even during the wet season, and they seem to have a relative maximum in March, at the end of the wet season. This is consistent with increased biogenic (such as fungal) activity with rain, high atmospheric moisture content and warm conditions during the summer. All this suggests that the biogenic sources from the nearby Amazonian basin have a continuous influence on this site throughout the year.

### 3.4 Special case: increased sulfate contribution after 2013

The increased concentration of $SO_4^{-2}$ in $PM_{2.5}$, $PM_{10}$-B and $PM_{10}$-C compared to $PM_{10}$-A (Table 2 and Table 5) calls for more in-depth analysis. In fact, $PM_{2.5}$ sampling (2013-2015) coincides with the onset of degassing volcanic activity in the upwind Arequipa and Moquegua regions in 2014 (Global Volcanism Program, 2023). Volcanoes are known sources of $SO_2$, which is transformed into $SO_4^{2-}$ in the atmosphere (Eatough et al., 1994), though they can also directly emit $SO_4^{2-}$ (Allen et al., 2002). This increase in regional volcanic activity is the main hypothesis for mass concentrations of sulfate in $PM_{2.5}$ being higher than

those of $PM_{10}$ (2011-2013 and 2016-2020, and Figure 2). Before the onset of volcanic degassing in the region, mean sulfate concentration in Chacaltaya was 451 ng m$^{-3}$, and after it started, Chacaltaya concentrations become > 900 ng m$^{-3}$ (Table 2). Frequent tropospheric volcanic activity has been documented in the region through ice-core analysis, though the sources were not identified at the time (de Angelis et al., 2003). Recently one source has been identified as the Sabancaya volcano, with sulfate being dominantly from volcanic origin in $PM_1$ when the station is under westerly air influence (Aliaga et al., 2021).

However, Ubinas volcano has also been in an active phase since 2006 (Moussallam et al., 2017 and references therein). Both volcanoes are located upwind of Chacaltaya, and therefore their emissions can arrive together when transport conditions are favorable (section 3.3.1).

**Table 5. Results of the two-sided Mann-Kendall test merging all sampling periods. Values statistically significant are in *italic***

| Species | Probable sources | p value |
|---|---|---|
| $SO_4^{2-}$ | Dominated by volcanism | *<0.05* |
| $NH_4^+$ | Emissions close to the ground neutralizing most sulfate | *<0.05* |
| $MeSO_3^-$ (control) | Phytoplankton emissions transformed in the atmosphere | 0.09 |
| EC (control) | Urban + biomass burning | 0.13 |

To assess the influence of Peruvian volcanism in the sulfate burden at the station, we group up all samples that have a west (W) or northwest (NW) origin based on ERA-5 backtrajectories and compare them with $SO_2$ emissions for Sabancaya (400 km away from Chacaltaya) and Ubinas (300 km), which are available on an annual basis (Carn et al., 2017; Fioletov et al., 2023). Chacaltaya $SO_4^{2-}$ data from filter sampling were added up, using years with at least nine months of data, to obtain the annual accumulated value and also the annual mean. The results are presented in Figure 6.



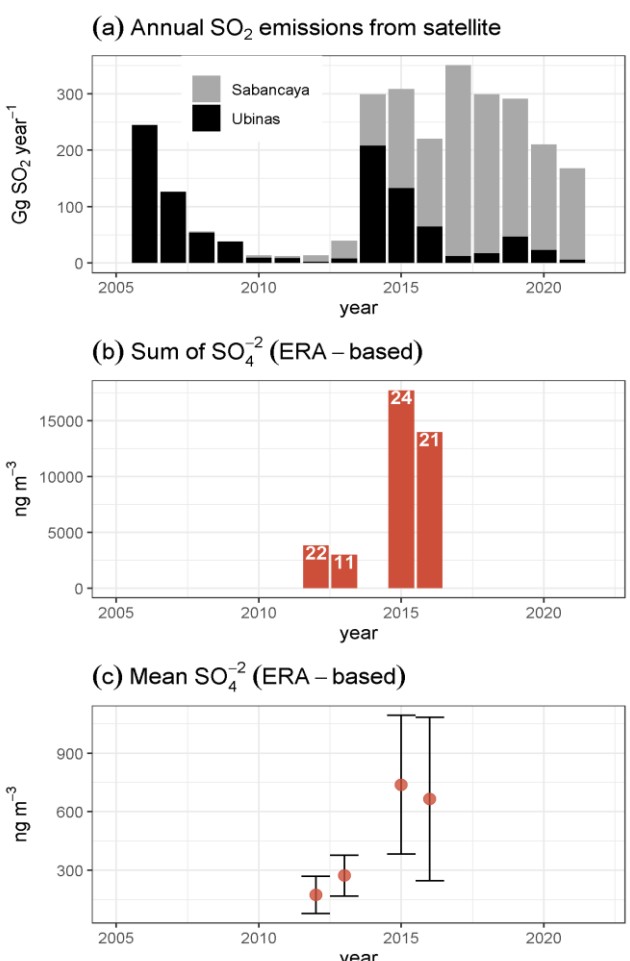

**Figure 6. Study case for volcanic transport. Only years with at least 9 months of filter data were used. (a) Annual SO₂ emissions for Sabancaya (grey) and Ubinas (black) from the global catalog of SO₂ emissions (Fioletov et al., 2023) (b) Accumulated SO₄⁻² measured in the filters taken under W and/or NW influence grouped per year based on ERA-5 backtrajectories. The number of samples used is in white letters inside the bars (c) Mean concentration and standard deviation for the selected samples.**

As observed from satellite detection (Figure 6a) the aforementioned volcanoes were in an active degassing phase since 2014 and the total sulfate detected at the station (Figure 6b) is in agreement with the sharp increase in $SO_2$ emissions. Even the mean sulfate concentration per sample (Figure 6c) is significantly higher after 2014. Note that 2012-2013 are statistically different from 2015-2016. As a control case, $MeSO_3^-$ transport was also calculated (Figure S10), indicating a relatively constant marine source with little circulation changes over the studied period. Moreover, as pointed out by Olson et. al. (2021), Sabancaya emissions do not seem to subside and influence nearby locations below the vent (volcanism contributes only to 9% of the measured sulfate in the city of Arequipa), which seems to be the case too at La Paz and El Alto, where the long-range secondary sulfate contributes only to 8% of the aerosol mass (Mardoñez et al., 2022). Therefore, we hypothesize that volcanic emissions are readily transported aloft, directly influencing high-altitude sites downwind such as Chacaltaya. In consequence, the





statistically significant increase in $SO_4^{-2}$ of our samples from 2014 onwards seems to be mostly related to an intensified
degassing process of Peruvian volcanoes leading to a regional increase in $H_2SO_4$.

### 3.5 Species transported under maximum/minimum influence of the convective planetary boundary layer to the station

A subset of $PM_{2.5}$ samples was studied with two objectives: first, to identify if there are differences between daytime and
nighttime concentrations of the measured species at Chacaltaya, and second, to identify if the species are transported from
nearby locations (or from long distances), all this in relation to the influence of the convective planetary boundary layer at the
station. These 20 samples span from November 2013 to December 2014 (gap from January to March 2014), and they were
obtained under maximum and minimum influence of the convective planetary boundary layer at the station. Samples obtained
under maximum PBL influence (*maxPBL*) correspond to 17 hours continuously sampled (09:00-02:00 BOT) than include the
diurnal development of the convective boundary layer and a nocturnal stable boundary layer that may be capped by a residual
layer (Collaud Coen et. al., 2018). Samples obtained under minimum PBL influence (*minPBL*) correspond to seven consecutive
hours that include nighttime and early morning, before the arrival of the convective boundary layer to Chacaltaya (02:00-09:00
BOT). This is a period when there can be an influence of a reminiscent residual layer but it also corresponds to the time when
we expect to capture more low-free-troposphere events.

Concentration and variability of concentration (represented by σ) for all species is higher during *maxPBL* than during *minPBL*
influence (Table 6).  However, only for EC, $NH_4^+$, $NO_3^-$ and glucose the significance level (α) is <10%. In practical terms,
most of the species present statistically similar concentrations for day and nighttime sampling, and this is the justification to
have pooled all types of samples (Table 2) in the previous sections of this work.

Nevertheless, the different significance levels reported (Table 6) present a structure that can provide hints for short- or long-
range transport.  With a significance level <10%, it is clear that for EC, $NH_4^+$, $NO_3^-$ and glucose transport under *maxPBL*
influence (mostly daytime and residual layer) dominates over the transport under *minPBL* influence (nighttime and early
morning). Therefore, this points to dominant sources nearby, in line with previous findings about the short-range influence of
urban pollution at the station (Wiedensohler et al., 2018; Bianchi et al., 2021). EC and $NO_3^-$ are typical traffic tracers that have
a marked diurnal cycle (Jorquera and Barraza, 2013; Bianchi et al., 2021; Mardoñez et al., 2022); and $NH_4^+$ and EC are
produced by litter or wood burning in the La Paz – El Alto urban area (Mardoñez et al., 2022). Ammonium can also be
produced by the decomposition of human and animal excreta, synthetic fertilizers, and biomass burning (Trebs et al., 2004;
Nowak et al., 2012; Behera et al., 2013) in nearby urban and sub-urban environments. Assigning a source below Chacaltaya
for these species agrees with $PM_1$ data obtained during an intensive campaign (Bianchi et al., 2021) where $NH_4^+$ and $NO_3^-$,
showed a diurnal cycle in Chacaltaya related to transport through the convective boundary layer. Finally, glucose is known to
present a marked daily cycle (Claeys et al., 2004; Wang et al., 2008) and its sources, as previously discussed, seem to be
dominated by emissions below the station (Altiplano, valleys) as fragmented vegetation and pollen from a radius of 200-300
km according to Brugger et. al. (2019).




For the rest of the species, we speculate that even if the defined significance level of 10% is not reached, we can infer an increased influence of long-range transport under *minPBL* conditions based on the varying degree of significance levels. A subgroup of species, namely SO$_4^{2-}$, OC, F$^-$, Cl$^-$, Br$^-$, C$_2$O$_4^{-2}$, K$^+$, Ca$^{+2}$ and mannitol, present a significance level between 14 and 35%. This may indicate that even if they tend to be transported under *maxPBL* (hence, with sources near the station), they may

not have very pronounced diurnal cycle and/or they may also be influenced by long-range transport. The complex transport mechanism described in section 3.3.2 may explain this. Finally, for Li$^+$, Na$^+$, MeSO$_3^-$, arabitol, Mg$^{+2}$ and levoglucosan, there is no significance level reached to find a difference between *maxPBL* and *minPBL*, their concentrations being statistically similar. This similarity points to a preferential long-range transport for those species that overwhelms local emissions. This is interesting because even if the Chacaltaya observatory is under stable atmospheric conditions between 02:00 and 09:00 BOT,

low free tropospheric intrusions are difficult to observe with high-volume sampling because they are not long enough (Scholz et. al 2023 reported free tropospheric episodes lasting less than 8 hours) to leave a marked fingerprint in our dataset. In this regard, high frequency (on-line) measurements of the aerosol chemical composition at the station prove their value as complementary to the long-term observations (Bianchi et al., 2021; Scholz et al., 2023; Zha et al., 2023).

**Table 6. Chemical composition for PM$_{2.5}$ samples obtained during maximum (Mixing/Boundary Layer + Residual Layer) and minimum (Residual Layer + Low Free Troposphere) influence of the atmospheric boundary layer at the station Mean ($\bar{x}$), median ($\tilde{x}$), standard deviation ($\sigma$) and number of samples above quantification limit (N) are presented. Levoglucosan * includes its stereoisomers. Two-tailed Wilcoxon Rank sum test results as α (significance level in %).**

| Ambient concentrations ng m$^{-3}$ | Maximum influence ABL (ML+RL) 0900 to 0200 of next day | | | | Minimum influence ABL (RL+LFT) 0200 to 0900 | | | | $\frac{\sigma_{maxPBL}}{\sigma_{minPBL}}$ | $\frac{\tilde{x}_{maxPBL}}{\tilde{x}_{minPBL}}$ | Significance level α (%) |
|---|---|---|---|---|---|---|---|---|---|---|---|
| | $\underline{x}$ | $\tilde{x}$ | $\sigma$ | N | $\underline{x}$ | $\tilde{x}$ | $\sigma$ | N | | | |
| NH$_4^+$ | 215 | 185 | (122) | 8 | 120 | 122 | (49.7) | 12 | 2.5 | 1.5 | 1.5 |
| NO$_3^-$ | 61.5 | 62.0 | (28.4) | 8 | 32.8 | 35.4 | (14.0) | 12 | 2.0 | 1.8 | 2.3 |
| Glucose | 1.78 | 1.36 | (0.94) | 7 | 1.12 | 1.03 | (0.41) | 11 | 2.3 | 1.3 | 5.6 |
| EC | 42.6 | 39.0 | (24.3) | 8 | 23.0 | 22.4 | (4.68) | 11 | 5.2 | 1.7 | 7.5 |
| SO$_4^{-2}$ | 824 | 655 | (615) | 8 | 520 | 490 | (262) | 12 | 2.3 | 1.3 | 14 |
| OC | 549 | 569 | (277) | 8 | 357 | 313 | (95.5) | 11 | 2.9 | 1.8 | 16 |
| Mannitol | 0.84 | 0.72 | (0.46) | 7 | 0.55 | 0.59 | (0.14) | 11 | 3.2 | 1.2 | 18 |
| Br$^-$ | 1.12 | 1.24 | (0.57) | 7 | 0.82 | 0.78 | (0.28) | 10 | 2.0 | 1.6 | 23 |
| F$^-$ | 1.86 | 1.86 | (1.55) | 8 | 0.97 | 0.71 | (0.54) | 11 | 2.8 | 2.6 | 24 |
| K$^+$ | 17.9 | 19.4 | (10.2) | 7 | 12.03 | 10.1 | (4.61) | 11 | 2.2 | 1.9 | 29 |
| Cl$^-$ | 8.10 | 8.36 | (5.05) | 6 | 4.84 | 3.51 | (2.46) | 7 | 2.1 | 2.4 | 30 |
| C$_2$O$_4^{-2}$ | 21.1 | 20.0 | (12.4) | 7 | 15.1 | 13.5 | (9.77) | 10 | 1.3 | 1.5 | 32 |
| Ca$^{+2}$ | 77.0 | 55.0 | (76.3) | 8 | 52.6 | 42.5 | (51.4) | 12 | 1.5 | 1.3 | 34 |
| Levoglucosan * | 6.79 | 4.37 | (6.51) | 8 | 4.1 | 3.81 | (2.73) | 12 | 2.4 | 1.1 | 47 |
| Mg$^{+2}$ | 6.31 | 4.67 | (4.46) | 7 | 4.74 | 3.96 | (2.92) | 11 | 1.5 | 1.2 | 48 |
| Arabitol | 0.92 | 0.81 | (0.61) | 8 | 0.73 | 0.78 | (0.22) | 11 | 2.8 | 1.0 | 60 |
| Li$^+$ | 0.02 | 0.01 | (0.02) | 7 | 0.01 | 0.01 | (0.01) | 10 | 1.8 | 1.1 | - |
| Na$^+$ | 13.1 | 10.1 | (10.3) | 8 | 13.4 | 12.4 | (8.16) | 12 | 1.3 | 0.8 | - |
| MeSO$_3^-$ | 4.65 | 3.30 | (4.45) | 8 | 3.64 | 3.27 | (1.87) | 12 | 2.4 | 1.0 | - |



## 4 Summary and conclusions

In this study, we present a unique long-term record of the chemical composition of aerosol sampled at the Chacaltaya station, the highest in the GAW network. The information is valuable in a region otherwise poorly characterized, documenting the variability of an essential climate variable in the tropical Andes.

Concentration levels of $PM_{10}$ range from 2.4 to 22.6 μg m$^{-3}$ STP confirming that the site is regularly influenced by nearby sources such as rural emissions and urban pollution, and that the air sampled at 5380 m a.s.l. is not purely free-tropospheric.

The station is representative of the regional background in a radius of 200-1600 km, with the distance from the site depending on the transport conditions. Loss of coarse particles along the way seems important.

The seasonality of several species has been described for the first time in the region, showing lowest concentrations for most species during the wet months (December to March) and maximum concentrations during the dry months (April to November). This is due to the marked seasonality of both the transport conditions and intensity of source emissions.

Between July and September (though extending from June to November), the high concentrations of an important number of species (EC, OC, $K^+$, $Br^-$, $F^-$, $Cl^-$, $NO_3^-$, formate, oxalate and levoglucosan) are related to agricultural practices, which include biomass burning. Biomass burning practices clearly influence regional aerosol composition, contrary to what was found at this site four decades ago, when deforestation (and subsequent land use change) was just beginning in Amazonia and therefore the Andean aerosol was only marginally influenced by biomass burning emissions.

This study also reports key organic compounds emitted by direct biogenic emissions. The sugar alcohols arabitol and mannitol are not strongly correlated, which marks a different behavior compared to other documented sites. Other biogenic sources ($K^+$) are also detected during the wet season, but additional studies are needed to fully understand the behavior of primary biogenic aerosols in the Amazonian and Andean regions. All this gains relevance in a period when the deforestation advances rapidly in the region, and therefore documentation on its effects on the aerosol chemical composition is needed.

The cities of La Paz and El Alto and the surrounding activities over the Altiplano clearly affect the aerosol chemical composition with EC, $NO_3^-$ and probably oxalate as traffic indicators; $NH_4^+$ from both urban and agricultural activities; and glucose likely related to vascular vegetation debris (grasslands, agriculture). Additionally, insoluble mineral matter represents an important fraction (33-56%) of the aerosol, originating from dry soils. All these are transported from below the station along with the development of the convective planetary boundary layer and are likely to have an impact at a wide scale high-
altitude in the region.

The OC/EC ratio is similar to rural sites, and does not have a marked seasonality, likely due to the permanent influence of long-range transport, and the constant aging processes that take place in this tropical altitude region all year round in spite of the change in circulation patterns and source activity.

Sustained volcanic degassing from Sabancaya and Ubinas is hypothesized to influence the regional background of sulfate
especially at high altitude, as emissions are directly vented out above 5000 masl, where conditions for long-range transport are met. Interestingly, as active volcanoes lie on the path between the Pacific Ocean and Chacaltaya, their emissions arrive along



with aged marine air masses. The recent unveiling of those volcanic sources can help reinterpret paleorecords, in the light of the current knowledge of transport conditions to the tropical Andes. These measurements constitute one of the first long-term observations of aerosol chemical composition at a high-altitude site in the tropical Southern Hemisphere. The high altitude poses numerous challenges when it comes to maintain sample collection and analysis. However, these efforts have yielded valuable insights into this specific region, allowing us to effectively track and comprehend the natural and anthropogenic influences at play.

**Contributions**

IM, FV, LT performed the fieldwork. FV generated the back-trajectory data and LT, MAK, DA contributed to their treatment. AA made possible the mass and elemental measurements; JLJ supervised the chemical analyses and provided most of the funding for doing them. IM, RK, GU, AA and PL conceptualized the manuscript. IM curated and analyzed the data, and prepared the manuscript with contribution from all co-authors.

**Open access data**

Data are available through the WMO/GAW WDCA monitoring database (EBAS) currently hosted at the Norwegian Institute for Air Research (NILU) at https://ebas.nilu.no/

**Competing interests**

R. Krecji is editor in ACP

**Acknowledgements**

We acknowledge all the personnel of IIF-UMSA for assisting with the station functioning (M. Agramont, Ing. P. Miranda), maintenance (N. Choque, E. Cusi, T. Zabaleta) and electric troubleshooting (F. Avila, I. Rivero) and the IRD (Institut de Recherche pour le Développement) delegation in Bolivia for assisting with logistics and customs clearance for the instruments. The instrumental deployment at Global GAW station used in this study is supported by an international consortium funded by IRD, Centre National de la Recherche Scientifique (CNRS under SNO-CLAP program) and Ministère de la Recherche (under ACTRIS-FR activities), Observatoire de Sciences de l'Univers de Grenoble (under Labex OSUG@2020), Leibniz Institute for Tropospheric Research, Consejo Superior de Investigaciones Científicas (CSIC), Laboratoire des Sciences du Climat et de l'Environnement (LSCE) and University of Stockholm.
Many people are to be thanked at IDAEA and IGE for the analyses of the overall series of filters, essentially non-permanent engineers from the IGE Air Quality Group (A. Waked, C. Charlet, F. Donaz, F. Masson, S. Ngo, V. Lucaire, A. Vella, C.



Verin, R. Elazzouzi), together with many students and trainees. Analytical aspects were supported at IGE by the Air-O-Sol
platform within Labex OSUG@2020 (ANR10 LABX56).

We thank Stefano Decesari (ISAC-CNR) for making chemistry of the Nepal Climate Observatory available, Danilo Custoio
(USP, HZH) for clarifying the details of Mato Grosso cities, and Yves Mousallam (Columbia University) for sharing
Sabancaya plume composition.

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
