# Peer review of "Tropical tropospheric aerosol sources and chemical composition observed at high-altitude in the Bolivian Andes"

_EGUsphere, 2023_

## Author Comment (AC1)

**Reply to referee # 1**

The authors would like to thank referee #1 for taking the time to review the manuscript. We are grateful for the comments and suggestions which allowed us to improve the manuscript. We followed their advice, and an additional consistency revision was made to the original text.

In the manuscript tracking the changes, modifications made to comply with referee #1's comments were highlighted in yellow, aside the replies stated in the next tables.

Specific comments:

| Nº | Comment | Reply |
|---|---|---|
| 1 | Several statements regarding potential sources of $PM_{10}$ and $PM_5$ are qualitative (Abstract, lines 34-40). A source apportionment (SA) should have been carried out to achieve quantitative conclusions about major sources impacting the monitoring site. Several manuscript authors have already done so for the closest urban area of La Paz – El Alto (Mardoñez et al, Source apportionment study on particulate air pollution in two high-altitude Bolivian cities: La Paz and El Alto, Atmospheric Chemistry and Physics Discussions, 1–41, https://doi.org/10.5194/acp-2022-780, 2022). A comparison of Chacaltaya SA results with those already published for the neighbor urban area would provide an in-depth quantitative analysis and would enhance the manuscript's scientific value. Most published SA studies present a chemical speciation campaign followed by application of a receptor model. | We have included the source apportionment of the Chacaltaya dataset made with EPA PMF v5.0.14 software. The high-altitude implies that the concentrations are pretty low, quite mixed during transport, and with a low range of concentrations, making variability low and co-linearity high. However,  the obtained results are statistically sound, even if single sources were not obtained (except for a classical biomass burning source), in part due to the insufficient number of species to constrain the solution, and in part due to the aforementioned characteristics of high altitude sampling.

We have added some sections about SA: section 2.7 in methods, and section 3.2 in results, and an extensive supplementary material. We use the SA results along the text to sustain the discussions. |
| 2 | In close connection with the above comment, the discussion in section 3.3 would benefit of presenting SA results beforehand, so seasonality would be discussed in terms of sources rather than by species (that may come from several sources). | SA results were included in the interpretation of section 3.3 |
| 3 | In Section 3.1, two estimates of OM/OC ratio were used, because of seasonality. Is it possible to estimate that ratio by linear regression of (PM-inorganic mass) against OC? This could be carried out by season to account for such variability. In this way, the uncertainty in OM would be reduced. | This is a valuable suggestion and we will take it into account for future works, but in this case we consider that we do not have enough information to perform a statistically sound regression. Indeed, we only have 5 samples with measured PM-inorganic mass simultaneously to OC.

* Please note that section 3.1 became section 3.5 after reorganization of the manuscript. |
| 4 | In section 3.2.2 (lines 363-376) it is discussed that OC/EC is ≈ 10 with little seasonality, and this is ascribed to long-range, aged aerosol dominates with a high SOA contribution to OC. I do not understand the hypothesis stated in lines 369-370: why is this hypothesis needed to explain these OC/EC ~ constant results? | The UV influence hypothesis is indeed not needed. For clarity, we have removed it from the paragraph. According to the SA, 29% of OC* has an urban origin. This confirms that the long-range transport dominates the OC* burden, and therefore this may be the reason why OC/EC presents little variability. The urban influence is not defining the seasonality of the OC/EC ratio. |
| 5 | Section 3.5: the discussion that ends with Table 6 would have improved with a SA result for Chacaltaya beforehand. | Section 3.5 was removed in agreement with the suggestion of referee # 2, but  table 6 was moved to the supplementary material.

* Please note that table 6 is table S7 in the revised version. |
| 6 | Conclusion section: I think there are contradictory statements here. First, on lines 630-631, it is mentioned that "La Paz and El Alto … activities… affect the aerosol chemical composition (at Chacaltaya) with EC, $NO_3$ … as traffic | We agree with the referee. The three paragraphs mentioned here were modified for clarity.

In the revised manuscript, the aforementioned modified lines are 344-351 and 626-635. |

| | |
|---|---|
| indicators… ”. Then, in lines 636-637 it is stated that "OC/EC ratio … does not have a marked seasonality … likely due the permanent influence of long-range transport". However, OC is also emitted by traffic, and it is mentioned that OC/EC ratios for La Paz – El Alto range between 2 – 3.5 (approx.). Then, I do not understand why EC from La Paz -El Alto would impact Chacaltaya but not OC emitted from the very same area — given that in lines 369-379 the authors hypothesized that "… the high UV of the tropical atmosphere over the Altiplano could play a role in the impressively fast aging of the organic matter at this site when transported from the nearby urban area." This issue needs to be clarified. | |

Technical corrections:

| Nº | Technical correction | Reply |
|---|---|---|
| 1 | I think figure S12 should be referred to instead of S10 (line 122). | Corrected |
| 2 | In Section 3.3, Figure 5 is hard to visualize. I would recommend splitting it in several graphs, perhaps moving some to supplementary information. | We have split figure 5 in three, corresponding now to figures 5, 6 and 7. |
| 3 | Since this is not the first report about Chacaltaya measurements, sections 2.1 and 2.2 could be shortened by moving some paragraphs to Supplementary Information. | Section 2.1 was shortened, but 2.2 was not easy to shorten as it needs to explain the complexity of the sampling at this site. |

---

## Author Comment (AC2)

**Reply to referee # 2**

The authors would like to thank the anonymous referee #2 for taking the time to review the manuscript. We are grateful for the comments and suggestions which allowed us to improve the manuscript. We followed the given advice, and an additional consistency revision was made to the original text.

In the document tracking the changes, modifications made to comply with referee #2's comments were highlighted in cyan, aside the replies stated in the next tables.

| Nº | Comment | Reply |
|---|---|---|
| 1 | It is not entirely clear how many new insights have resulted from considering the full 2011-2020 record now available, compared to findings already published from shorter campaigns at the same site. This issue should be addressed in a revised and partly reorganized version, that also seeks to clarify some confusion in sections of the current draft that will be described below. | The reorganized manuscript addresses this comment.

Briefly, we wish to highlight that this work:
- presents results of $PM_{10}$ and $PM_{2.5}$ aerosol mass and chemical composition of the aerosol (ions, EC OC, anhydrosugars), which were not previously available for this region as other studies include other chemical species.
- integrates many already published works in a comprehensive interpretation of the observed seasonal cycles |
| 2 | To me, the most significant example of this problem is section 3.5, which seeks to ascribe small (largely not significant) differences between the concentrations observed during the daytime versus nighttime to boundary layer dynamics. Issues that strike me are that

1) all earlier subsections of Results and Discussion combine daytime, nighttime, and 24-hour filter samples precisely because any diurnal effects are so small,

2) details of the sampling protocol in this study seem to indicate that there were large temporal offsets between any given pair of day/night samples that will complicate comparisons and then averaging 6-8 daytime and 7-12 nighttime samples before comparing would seem to combine multiple possible factors controlling concentrations and likely obscure any impact dominated by boundary layer variations, (more on this later), and

3) several other short term studies that made faster measurements are cited that provide stronger evidence of vertical mixing bringing local pollution to the site for relatively short episodes.

I recommend that Section 3.5 be removed from the revised version. | We agree with the reviewer, and section 3.5 was removed from the revised version of the paper

1 ) In order to justify the merging of all sampling periods, table 4 was moved to the supplementary material (it is now table S7) because it provides the statistical evidence for combining daytime, nighttime, and 24-hour filter samples.

2) and 3) Agreed, though most short-term studies focus on $PM_1$, and this study is mostly devoted to $PM_{10}$. |
| 3 | 30-32 I found it disconcerting to read in the abstract that concentrations in PM2.5 tended to be higher than in PM10. This should not be possible for samples collected simultaneously. Much later (line 161 and in Table 1) it becomes clear that sampling was either behind PM 10 or PM 2.5 impactors but not both at same time. The inference that most of the measured compounds were dominantly on smaller particles, based on the similar concentrations measured in PM 2.5 and PM 10 samples, seems well founded. However, the abstract needs to mention that the PM 2.5 and PM 10 sampling occurred during non-overlapping periods, and might point out that it seems most aerosol mass is found on the PM 2.5 fraction. | Thank you for pointing out this.
The term "non-overlapping" was added in the first line of the abstract, and line 34 was modified in agreement with this recommendation. |

| Nº | Comment | Reply |
|---|---|---|
| 4 | 40-41 A mean (or median) of 10.5 with standard deviation of 38.9 does not seem consistent with the statement that the EC/OC ratio was "practically constant" year-round. | Thank you for pointing out this. Standard deviation is indeed 5.7. The value 38.9 was obtained from the raw dataset, which was not the case of the rest of the manuscript. Three cases of OC/EC ratios are anomalously high because they are driven by EC values close to the detection limit (Samples 38,51,192). These values were removed from the calculations for the "Results and discussion" section but the data in the abstract was not updated for the first version of the manuscript. The correct value of the standard deviation 5.7, and it is now in the abstract. We have also added the interquartile range values for clarity. |
| 5 | 95-97 Are La Paz and El Alto 2 different cities that are close to each other, or is one specifically a city and the other the surrounding metropolitan area that includes the city (like Los Angeles and the Los Angeles basin)? At the start of this sentence I thought El Alto was a city in the La Paz metro area because the population in 1976 was much larger in La Paz. But then it says that El Alto population in 2012 was much larger than the La Paz population (not possible if El Alto is part of La Paz). | La Paz and El Alto are two different cities, which have separate administrations. El Alto is located in the Altiplano plateau and La Paz in the valley. They share a physical border: the abrupt transition from the plateau to the valley. Therefore, and for clarification, the term "conurbation" was used instead of "metropolitan area" in the article. |
| 6 | 105 Since you note that much of the precipitation is solid you should clearly state that 865 mm annual average is water equivalent depth, or separately report the depth of rain and snow if necessary. | "Water equivalent depth" clearly stated in what is now line 104. |
| 7 | 108-109 "Wet-to- transition" should be "Dry-to-wet transition" | Corrected |
| 8 | 110-111 It does not seem to make sense for winds coming to the station from the SE and E to be channeled through valleys N of the station. Could work for NE winds. | Atmospheric circulation in this mountain region is quite complex. The statement comes from an already published study using the Flexpart dispersion model (Aliaga et al. 2021). The lines 109-113 were modified for clarity. |
| 9 | 117-125 This paragraph is part of the reason I suggest deleting section 3.5. Findings based on the long data set in that section are kind of weak, and largely already established | Section 3.5 was deleted. |
| 10 | 145-155 In combination with Table 1, this paragraph claims to provide details of the sampling schedule. However, I feel some important details are missing. In particular, it is not clear how you alternated between day, night, and 24-hour samples. If, for example, you collected 8 day, then 7 night, then 2 24-hour samples during the wet season in PM10-A, the mid points of the first day and last night samples would have been separated by ~42 – 98 days. This would make it hard to ascribe any differences in concentrations solely to boundary layer dynamics. Even if you alternated between day and night samples the temporal offset poses a challenge in terms of attributing differences to specific process(es).

Not sure whether it would be easier to describe the actual sampling schedule in the text or in Table 1. | The samples were set to be obtained in an alternating sequence like: day-night-day-night-blank.

Lines 144-146 were modified to make the procedure more understandable without modifying table 1.

We agree with the referee that the temporal offset poses a challenge in terms of attributing differences to specific process(es) and we are aware of the limitations of our sampling protocol. |

| Nº | Comment | Reply |
|---|---|---|
| 11 | 198 "elementary" should be "elemental" | Corrected |
| 12 | 218-219 Why was preservation of fluoride, chloride, and nitrate not a problem in periods A and B? Related question, how confident should the reader be in the fluoride, chloride, and nitrate concentrations in the earlier periods, including for PM 2.5? | The sample shipping was done on a more regular basis for $PM_{2.5}$, $PM_{10}$A and B samples, and therefore the analyses were made faster than for batch $PM_{10}$C. Batch $PM_{10}$C could not be shipped within a year of the sampling because of Covid-19 lockdowns. Moreover, as the university campus remained closed for several months, we have no information on possible power outages or other possible problems that may have arisen during those additional storage months. |
| 13 | 237 HYSPLIT is Hybrid Single-Particle Integrated Trajectory (add underlined parts) | Corrected |
| 14 | 262-263 Consider providing more details on the comparison between trajectories driven by WRF versus ERA-5. The short statement here suggests that the ones with ERA are suspect, yet they are used for most analyses. | We are aware that the WRF (1-km) backtrajectories do a better job than the ERA-5 (30 km) backtrajectories, based on sample to sample comparison, and using eBC and CO from other studies. However, we used ERA-5 backtrajectories in this work because of their disponibility. This was clarified in the text. The modified paragraph corresponds now to lines 232-251 |
| 15 | 265-273 This paragraph is jumpy, making it hard to know what you are trying to emphasize. I also note that it is not customary to refer to Table 4 prior to mentioning Tables 2 and 3, then Table 5 also before Table 3. In addition, it is not clear what hypothesis is tested with the Mann-Kendall test (Table 5). Here it is suggested the test was whether concentrations day/night were different, Section 3.5 suggests you were comparing across the 4 sampling periods, and the words in the table suggest somehow this test allowed source attribution. | Thank you for pointing out this. The paragraph (lines 265-274) was simplified to make it more understandable. |
| 16 | 271 Is it surprising or notable that the very low concentrations during wet season were the most statistically similar? | It is not, because as stated in lines 273-274, wet soils may prevent dust remobilization and other processes may take place during the wet months. |
| 17 | 278-296 I do not understand the decision to begin Results and Discussion with Section 3.1 which focus on a very small subset of data. Would seem better to start with 3.2 and 3.3 and come back to this as another "special case" before or after Section 3.4. | Thank you for your suggestion, Section 3.1 was moved and it became Section 3.5 in the revised manuscript. |
| 18 | Table 3. Suggest reporting TE in micrograms/m^3 like everything else. | Unit changed to $\mu g\ m^{-3}$ |
| 19 | 304-306 Would be helpful to remind reader here that PM 10 and PM 2.5 samples were collected at different times. | A sentence clarifying this was added in the first line of the paragraph. Line 281 in the revised manuscript reads now: "$PM_{10}$ and $PM_{2.5}$ samples were collected during non-overlapping periods. " |
| 20 | 315 seems either there are words missing after "and" or that "and" should be deleted in "nitrate and stands" | "And" word removed. |

| Nº | Comment | Reply |
|---|---|---|
| 21 | 337-339 This summary of prior work on sulfate sources at Chacaltaya is part of the reason I said it is not so clear what the new long data set brings to the story. | Previous works (Bianchi et al 2021, Aliaga et al 2021, and Scholz et al 2023) only focus on the period of the SALTENA campaign (December 2017 to June 2018) when the Sabancaya volcano was the dominant source of SO2, emitting approximately 300 Gg of SO2 per year. Our multi-year study actually measures sulfate from two very different volcanic emission regimes. First, in 2012/13 when there were very few emissions from either volcano (totalling around 30 Gg of SO2 per year), and secondly in 2015/16 when the emissions had increased tenfold. Therefore, the long dataset provided in this study is novel and hints that volcanoes are indeed the main source of SO2 and sulfate sampled at CHC. |
| 22 | 363-365 Two questions here. What about April, it is notable by not being included in any of the seasons?

And how can the standard deviation be 38.9 over the entire study (line 40-41) but never higher than 7 in any season. | April was not included because it represents a single month with high year-to-year variability. However, it was now added, and it also fits the range of variability of a "rural" site.

Standard deviation is indeed 5.7 as explained in the answer Nº4. |
| 23 | 369-370 It does not seem that the OC/EC ratio provides strong evidence for "impressively fast aging" so not sure why this sentence is inserted in this paragraph. | Also in agreement with referee #1's observation, this paragraph was modified. Lines 350-351 were added. |
| 24 | 378-385 This is nice background information, but what is interesting or notable about the Chacaltaya results in Fig. 3? | This background information is intended to provide the reader with basic information of the use of each primary biogenic aerosol tracer. However, it was shortened for clarity. |
| 25 | Figure 3 might work better if the scaling on Y axis was same in both panels. | The same scale was set for both panels of figure 3. In addition, the bar plot was slightly modified as it was observed that R is not able to correctly handle stacked bars in a logarithmic scale. |
| 26 | 403 "does not to have" should be "does not" or "does not seem to" | Changed for "does not seem to" |
| 27 | 414-415 Does it make sense to aggregate all 4 sampling periods to assess seasonality? In particular I wonder about sulfate, which you later show has a step change due to volcanic emissions that may obscure seasonality.

It might make sense to check seasonality in each period separately. Might not want/need to show these results, but you could note whether the seasonality is persistent across the study (and perhaps focus on species for which it is not to see if there is information there). | We aggregated all 4 sampling periods to increase the statistical robustness of the calculations, but we present the data of each period in figure 5 following a color scale for each sampling period. In figure 5, period $PM_{10}A$ (sulfate panel, black dots) can be observed to fit the same seasonality of the other periods.

For the wet season fewer samples were collected than for the dry season (as seen in Fig. 4). If we had worked with separate periods we would have remained with months with less than 3 samples, losing statistical power. However, data segregated by sampling period is presented in a spreadsheet in the supplementary material. |
| 28 | 424 In section 2.1 you seemed to imply that westerly winds were quite rare, so a little surprising to hear there is a season with significant westerly flow. | Thank you for pointing out this. Winds with a westerly component are indeed quite frequent (Chauvigné et al 2019, Aliaga et al. 2021). In section 2.1, line 113, this was clarified: "In the dry and dry-to-wet seasons, winds with a westerly component blow over the Altiplano towards the station" |

| Nº | Comment | Reply |
|---|---|---|
| 29 | 435-440 Interesting that you find significant marine-sourced MeSO3^-, but suggest that nearly all sodium and magnesium are crustal. One might expect some sea-salt with the MeSO3^-. Might be worth looking at case studies rather than the monthly averages. | We agree with this comment. We have added lines 453-456 about the assignment of $Na^+$, $Mg^{2+}$, $Ca^{2+}$, $K^+$ to a marine contribution based on source apportionment results (now included in the manuscript as sections 2.7 and 3.2 in compliance with referee #1's suggestions). |
| 30 | 445-447 This sentence almost contradicts the one I pointed to immediately above. | The source apportionment helps clarify this. The paragraph has been reorganized (now lines 448-457). |
| 31 | Section 3.3.2 It is quite surprising to me, and may be to others, that you identify a biomass burning (BB)cluster that does not include ammonium. You may want to confront this in this section, rather than just noting that ammonium peaks in the dry season, often correlated with sulfate, and coming back to it in section 3.5 which I suggest be deleted. | Source apportionment indicates that the BB contribution to $NH_4^+$ without association to $SO_4^{2-}$ was around 24% (biomass burning 5%+ combustion/urban 19%). This may be only partially true because the origin of the ammonium associated with sulfate was not identified. However, we clarify this in lines 501-506 of the revised version. |
| 32 | 460 Not sure "notorious" is the correct word here. | "Notorious" word removed |
| 33 | 485-510 Text here seems to muddle your story. I grant that most of the things measured have more than a single source, but this section is supposed to be focused on JAS when smoke seems a significant if not dominant source. My point is why would possible marine, urban, volcanic sources contribute to peaks in selected compounds in late summer, but other compounds that also come from some of these sources do not show significant enhancements. | We have modified most of this paragraph for the sake of clarity. In its place (now lines 500-513) we have included a paragraph including source apportionment results related to the biomass burning group. Some of the statements made previously fit in this new paragraph, others were moved to more appropriate sections or just eliminated. This helps to some extent to disentangle some of the other suspected sources. |
| 34 | 515-516 If lithium is often near detection limits, why focus on it? And why suggest it may come from BB in SON when previous section points to JAS peak in BB influence? | The complete section 3.3.3 was improved. In spite of low values, $Li^+$ seems to present maxima in the late BB season. We have, however, simplified to the maximum the statements about Li in now lines 532-534 It needs to be clarified that June to November encompasses the BB season, and this was modified in section 3.3 to avoid oversimplification of JAS being the only period of biomass burning influence. |
| 35 | 520-523 Speculation about glucose, mannitol, and ararbitol seems weak. Why would high variability indicate continuous influence from the Amazon. The March peaks are not striking in Fig 5, in fact all seem enhanced in Aug-Nov nearly as much as in March. | This paragraph (now lines 520-528) was modified including parts of section 3.5 and source apportionment results. |
| 36 | Section 3.4

How does the proposed increase in volcanic emissions after period A fit with the earlier finding that sulfate peaks in dry season (section 3.3.1)? Seems unlikely that the volcanoes track seasons.

Would the seasonal variation of sulfate change if you removed samples with W or NW trajectories before calculating monthly averages? Main point is that different sections of this manuscript need to be somehow connected. | \* Please note that Section 3.4 is now section 3.5 in the revised manuscript.

It is true that volcanic emissions can be observed at any time of the year given favorable transport conditions (Aliaga et al. 2021), and that is the reason why $SO_4^{2-}$ is also present during the wet months at the station. However, If we remove the samples containing W or NW trajectories we would remain with years (2014, 2016, 2017) without any samples at all during the dry season. Therefore, it is difficult to apply this suggestion to the dataset.

Nevertheless, efforts were made to better |

| Nº | Comment | Reply |
|---|---|---|
| | | interconnect the sections of the manuscript. In this regard, we have moved to this section the information about F- , Cl-, Br- that was before in section 3.3.2 (now in lines 546-551). |
| 37 | Fig 6. Why show the volcanic emissions from 2005 through 2011 (before you have aerosol measurements) in these plots? | The scale of the x-axis was modified by removing 2005-2011 data. |
| 38 | Section 3.5. No detailed comments given recommendation that entire section should be deleted. | Section deleted. Explanations about $NH_4^+$, $NO_3^-$ from this section were moved to sections 3.1 and 3.3 |
| 39 | 636-638 Confusing to claim important year-round influence of long-range transport immediately after emphasizing local sources. | The source apportionment study suggested by referee #1 (new sections 2.7 and 3.2) helped estimate the nearby urban (29% of OC) and long-range (71%) influences to this site. The lines 628-637 were modified accordingly. |